

# *Fifty-six years of Surface Solar Radiation and Sunshine Duration at the Surface in São Paulo, Brazil: 1961 - 2016*

**Marcia Akemi Yamasoe[1], Nilton Manuel Évora do Rosário[2], Samantha Novaes Santos Martins Almeida [3], Martin Wild[4]**

[1] Departamento de Ciências Atmosféricas, Instituto de Astronomia, Geofísica e Ciências Atmosféricas, Universidade de São Paulo, São Paulo, Brazil

 [2] Departamento de Ciências Ambientais, Universidade Federal de São Paulo, Diadema, São Paulo, Brazil

[3] Seção de Serviços Meteorológicos do Instituto de Astronomia, Geofísica e Ciências Atmosféricas, Universidade de São Paulo, São Paulo, Brazil

[4] Institute for Atmospheric and Climate Science, ETH Zurich, Switzerland

Correspondence to: M. A. Yamasoe (marcia.yamasoe@iag.usp.br)



## *Abstract*

Fifty-six years (1961 – 2016) of daily surface downward solar irradiation,
sunshine duration, diurnal temperature range and the fraction of the sky covered by
clouds in the city of São Paulo, Brazil, were analyzed. The main purpose was to
contribute to the characterization and understanding of the dimming and brightening
effects on solar global radiation in this part of South America. As observed in most of
the previous studies worldwide, in this study, during the period between 1961 up to the
early 1980's, more specifically up to 1983, a negative trend in surface solar irradiation
was detected in São Paulo, characterizing the occurrence of a dimming effect. A similar
behavior, a negative trend, was also observed for sunshine duration and the diurnal
temperature range, the three variables in opposition to the trend in the sky cover
fraction. However, a brightening effect, as observed in western industrialized countries
in more recent years, was not observed. Instead, for surface downward irradiation, the
negative trend persisted and still in consonance to the cloud cover fraction increasing
trend. The trends for sunshine duration and the diurnal temperature range, by contrast,
changed signal. Some possible causes for the discrepancy were discussed, such as the
frequency of fog occurrence, urban heat island effects, aerosol changes and greenhouse
gas concentration increase. Future studies on aerosol effect are encouraged, particularly
with higher temporal resolution as well as modeling studies, to better analyze the
contribution of each possible causes.



## 1   Introduction

Ultimately, the downward solar radiation at the surface is the main source of energy that drives Earth's biological, chemical and physical processes (Wild et al., 2013, Kren et al., 2017), from local to global scales. Therefore, the assessment of the variability of the downward solar radiation at the surface is a key step in the efforts to understand Earth's climate system variability. Before reaching the surface, solar radiation can be attenuated mainly by aerosols and clouds, through scattering and absorption processes, and to a lesser extent, through Rayleigh scattering by atmospheric gases, absorption by ozone and water vapor, for example. In this context, during the last half-century, long term changes in the amount of surface solar radiation (SSR) have been investigated worldwide (Dutton et al., 1991, Stanhill and Cohen 2001, Wild et al. 2005, Shi et al., 2008, Wild, 2009, 2012, Ohvril, et al., 2009). At least two trends have been well established and documented, a decline in surface solar radiation between 1950s and 1980s, named *"Global Dimming"* and an increase, from 1980s to 2000s, termed *"Brightening"* (Stanhill and Cohen, 2001; Wild, 2009, 2012).

The global dimming definition, according to Stanhill and Cohen (2001), refers to a widespread and significant reduction in global irradiance, that is the flux of solar radiation reaching the earth's surface both in the direct solar beam and in the diffuse radiation scattered by the sky and clouds. However, among these studies, while the dimming phase has been a consensus for all locations analyzed, the brightening phase was not (Wild, 2012). Over India, for example, the dimming phase seems to last throughout the 2000s (Kumari and Goswami, 2010). The continuous dimming in India and the renewed dimming in China from 2000s, opposing to a persistent brightening over Europe and the United States, have been linked to trends in atmospheric anthropogenic aerosol loadings (Wild, 2012). By contrast, other studies suggested that



changes in cloud cover rather than anthropogenic aerosol emissions played a major role
in determining solar dimming and brightening during the last half century (Stanhill et
al., 2014). Therefore, the drivers of dimming and brightening are a matter of ongoing
research and debate. The role of these trends in the masking of temperature increase due
to the greenhouse gases (GHG) has been discussed (Wild et al., 2007). Furthermore, a
comprehensive assessment of the spatial scale of both dimming and brightening is
critical for a conclusive analysis of the likely drivers and implications for the current
global climate variability. Large portions of the globe are still lacking any evaluation on
this matter, such as Africa (Wild, 2009), which is a challenge for the spatial
characterization of both dimming and brightening trends.
Among the rare studies focusing on the South American subcontinent, Raichijk
(2012) discussed the trends over South America, analyzing sunshine duration (SD) data
from 1961 to 2004. The author divided South America in five climatic regions. In three
of them, also the one where the city of São Paulo is located, statistically significant
negative trends were observed on an annual basis, from 1961 up to 1990. From 1991 to
2004 a positive trend was observed in four of the five regions with a significance level
higher than 90%.
The alternative use of SD is mainly due to the lack of a consistent long-term
network for the monitoring of SSR across the continent, therefore alternative proxies
have to be found in order to provide an estimate of SSR long term trends. Another
variable commonly used to investigate SSR trends is the diurnal temperature range
(DTR), the difference between daily maximum ($T_{max}$) and minimum ($T_{min}$) air
temperature measured near the surface (Bristow and Campbell, 1984, Wild et al. 2007,
Makowski et al. 2008).



The present study takes advantage of fifty-six years of a unique high quality
concurrent records of surface solar irradiation (SSR), sunshine duration (SD), diurnal
temperature range (DTR) and sky cover fraction (SCF), i.e., the fraction of the sky
covered by clouds, from 1961 to 2016, in the city of São Paulo, Brazil, to provide a
perspective on dimming and brightening trends with an extended database.
Thus, we propose to answer two questions in this study: 1) How was the decadal
variability of SSR over the 56 years of data?; 2) Can SD and DTR be adopted as proxies
to infer SSR variability in São Paulo? To answer to these questions, we organize the
manuscript as follows: in part 2 we present the data and methods of analysis; section 3
is divided in 3 parts. In the first part we discuss the annual trends in SSR, SD and DTR;
in the second, we focus the analysis on cloud free days; in the third part of section 3 we
discuss the trends in the maximum and minimum air temperatures near the surface.
Section 4 summarizes the main conclusions and discusses possible future work on the
subject.

## 2    Observational Data and Methods

The long term measurements used in this study were collected at the
meteorological station operated by the Instituto de Astronomia, Geofísica e Ciências
Atmosféricas from the Universidade de São Paulo (IAG/USP), located at latitude
23.65º S and longitude 46.62º W, 799 m above sea level. Figure 1 shows the
geographical location of the meteorological station. The site is surrounded by a
vegetated area due to its location inside a park.



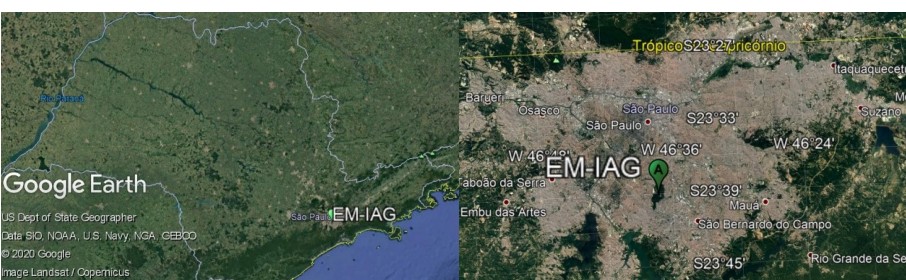


Figure 1 – São Paulo state and a zooming in view of São Paulo Metropolitan Area and
the location of the meteorological station of Instituto de Astronomia, Geofísica e
Ciências Atmosféricas from Universidade de São Paulo (EM-IAG). Adapted from ©
Google Earth (US Dept. of State Geographer – Data SIO, NOAA, U. S. Navy, NGA,
GEBCO - Image Landsat/Copernicus).

The downward solar irradiation has been measured since 1961 using an

*Actinograph Fuess model 58d*, with 5% uncertainty (Plana-Fattori and Ceballos, 1988).
Sunshine duration data was collected with a Campbell-Stokes sunshine recorder
(Horseman et al., 2008) from 1933 to the present, while daily maximum and minimum
air temperatures started to be monitored in 1935. Daily maximum and minimum
temperatures were used to estimate the diurnal temperature range as it is simply the
difference between the maximum and minimum daily temperatures. Diurnal sky cover
fraction was determined from visual inspection made every hour from 7:00 AM to
6:00 PM (local time) (Yamasoe et al. 2017).

Annual mean values of downward solar irradiation data at the surface were used

to characterize dimming and brightening trends while sunshine duration and diurnal
temperature range measurements at the same site were used to provide independent
information.

In order to detect possible temporal changes, avoiding autocorrelation in the

data, the modified Mann-Kendall trend test proposed by Hamed and Rao (1998) was
applied to the variables, while the regression coefficient was estimated based on Sen



(1968). A statistically significant trend at the 95% confidence level was detected if the
absolute value of Z was above 1.96.

According to the meteorological station records, completely cloud free days are

extremely rare in São Paulo, being more common from June to the beginning of
September, corresponding to the southern hemisphere winter time, when dry conditions
prevail in the region (Yamasoe et al., 2017). The number of days without clouds per
year, from sunrise to sunset, varied from 1 to 23. Also, the impact of aerosol in SSR is
higher from August to October, when advection of smoke plume from long range
transport can reach São Paulo, summing up to the local pollution. Thus, in order to
analyze how clear sky conditions varied during the last 56 years, we restricted the data
to the months of July to October, to minimize the effect of any possible seasonal drift in
the aerosol characteristics throughout the years. Following Manara et al. (2016), days
with SCF of up to 0.1 were allowed, in order to increase the number of clear days per
year. Thus, only years with 9 or more days, in the specified months, were included in
the study.

For the analysis, atmospheric transmittance was estimated dividing the measured

daily surface irradiation (SSR) by the expected irradiation at the top of the atmosphere
(TSR). Daily measured sunshine duration (SD or n) was also normalized to the day-
length (N).  Top of the atmosphere irradiation and day length were estimated using
formulas proposed by Paltridge and Platt (1976).  Observers at the meteorological
station also take note on the occurrence of fog every day. If fog was observed, the day
received the number 1, otherwise, the number is 0. For each clear sky day, information
on fog observation was verified. The fraction of cloud free days with foggy conditions
for each year was then estimated for the months of July to October, to verify any
possible influence on SSR and SD. Moreover, since horizontal visibility information is





also registered at the same time as the sky cover fraction, we included this information
in this analysis as well. Table 1 presents the registered code for horizontal visibility and
the corresponding distance range. Horizontal visibility can also be affected by haze and
fog conditions but is less sensitive to cloud variability.

Table 1 – Adopted codes for visibility records at the meteorological station and
corresponding distance ranges.

| Code | Distance (meter) |
|---|---|
| 0 | Less than 50 |
| 1 | 50 to 200 |
| 2 | 200 to 500 |
| 3 | 500 to 1000 |
| 4 | 1000 to 2000 |
| 5 | 2000 to 4000 |
| 6 | 4000 to 10000 |
| 7 | 10000 to 20000 |
| 8 | 20000 to 50000 |
| 9 | > 50000 |




To complement the analysis, aerosol columnar loading information from satellite

products such as the Absorbing Aerosol Index (AAI) from multi-sensor retrievals
(TOMS, GOME-1, SCIAMACHY, OMI, GOME-2A and GOME-2B) (Herman et al.,
1997, Torres et al., 1998, Graaf et al., 2005, Tilstra et al., 2014) and aerosol optical
depth (AOD) from MODIS (Moderate Resolution Imaging Spectroradiometer) onboard
Terra and Aqua satellites (Kaufman et al., 1997) were included. Shortly, the Absorbing
Aerosol Index indicates the presence of aerosol particles in the atmosphere with high
absorption efficiency in the ultraviolet spectrum. The product analyzed is the annual



mean value with a spatial resolution of 1º by 1º in a box from 47º W to 46º W and 24º S
to 23º S which includes São Paulo Metropolitan Area, for the months of July to
October, from 1979 up to 2016 (http://www.temis.nl/airpollution/absaai/). The AOD
product is a combination of the Dark Target (Kaufman et al., 1997, Remer et al., 2005)
and the Deep Blue (Hsu et al., 2014) retrieval algorithms also degraded to the spatial
resolution of 1º by 1º, averaged annually from 2000 (for Terra) and 2002 (for Aqua) to
2016, also considering only the dry season months obtained from the NASA Giovanni
dataset site (https://giovanni.gsfc.nasa.gov/giovanni/).

## 185    3    Results

### 186    3.1    SSR, SD, DTR and SCF annual mean variability and trends

Figure 2 illustrates the time series of the annual mean values for SSR, SD, DTR

and SCF, showing that all the analyzed variables exhibited a large variability from year
to year. SSR, SD and DTR presented a decaying trend up to the beginning of the
1980's, in opposition, therefore consistent, to the SCF trend. According to Rosas et al.
(2019), who analyzed the same cloud fraction database from the meteorological station,
focusing on the climatology for different cloud types and base heights, all cloud types,
except for middle level clouds, presented a positive trend, which is confirmed by this
study. A statistically significant trend, at the 95% level, was observed for stratiform
cloud fraction of 4.8 % per decade and for cirrus of 1.4 % per decade, from 1958 to

1988.


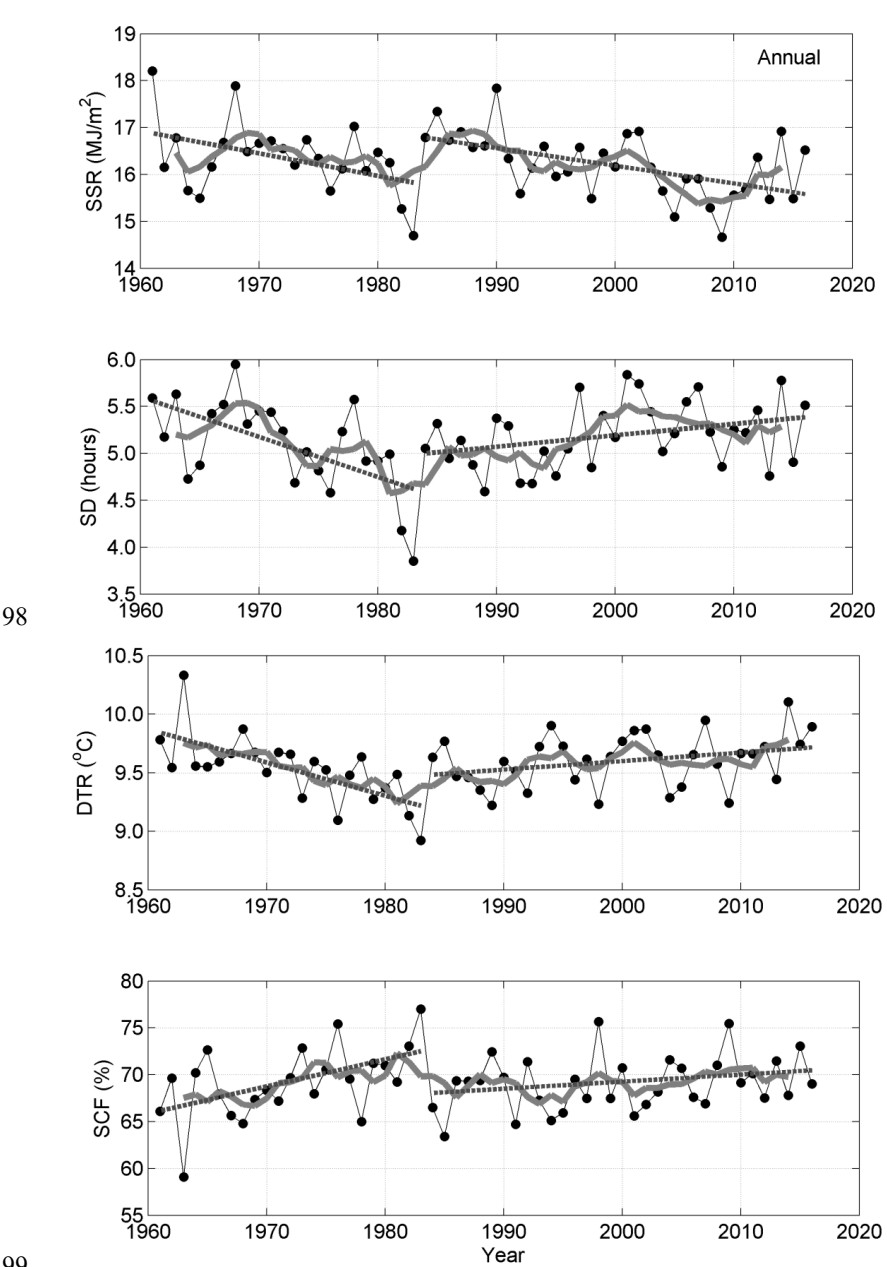



Figure 2 – Annual mean variability of surface solar irradiation (SSR), sunshine duration
(SD), diurnal temperature range (DTR) and sky cover fraction (SCF). Gray curves
represent 5 years moving averages and dotted lines are the result of trend analysis from
1961 to 1983 and from 1984 to 2016.



Returning to Figure 2, the gray curve represents the 5 years moving average,
while the dotted line indicates the result of the modified Mann-Kendall trend analysis,
discussed ahead. The year of 1983 was the one presenting the lowest annual mean value
for SSR, SD and DTR, clearly as a response to the peak of SCF observed in that year,
which is worth to mention, was characterized by a strong El Niño event. According to
the Earth System Research Laboratory from the National Oceanic and Atmospheric
Administration (ESRL/NOAA), it is listed amongst the 24 strongest El Niño events and
lasted from April 1982 up to September 1983
(https://www.esrl.noaa.gov/psd/enso/climaterisks/years/top24enso.html). This 1983 El
Niño effect was also detected in rainfall data over the São Paulo Metropolitan Area
(Obregón et al., 2014), although the authors claim that such influence, at least on
rainfall variability, is detectable but is multifaceted and depends on the life cycle of
each ENSO event.  Xavier et al. (1995), trying to identify a possible influence of ENSO
on precipitation extremes in the month of May, classified both May 1983 and May 1987
as exceptional extremes of precipitation. Their conclusion was that strong El Niño
events can affect the spatial organization of rainfall around São Paulo city. A more
recent study performed by Coelho et al. (2017), using daily precipitation data from 1934
to 2013 from the same meteorological station analyzed in this research, concluded that
El Niño conditions in July tend to increase precipitation in the following spring, also
anticipating the onset of the rainy season. No study was found about the possible effect
of ENSO on cloud cover over São Paulo. According to Rosas et al. (2019), middle and
high level clouds presented high positive anomalous cloud amount in 1983.
After 1983, the trend behavior of all variables changed, what motivated us to
separate the time series analysis in two periods, the first from 1961 to 1983 and the
second from 1984 up to 2016. The results of the modified Mann-Kendall trend test for



each period are presented in Table 2, considering both annual and seasonal variabilities.
Bold values indicate trends that are statistically significant at the 95% confidence level.
From the table, in the first period, SSR, SD and DTR presented a decreasing trend,
while SCF a positive one, increasing at a rate of 2.9% per decade. Except for SSR, all
trends were statistically significant, with daily SD decreasing at a rate of 0.37 hours per
decade and the diurnal temperature range declining at a rate of 0.49°C per decade.
Looking at the seasonal variability, southern hemisphere autumn (MAM) and winter
(JJA) presented statistically significant decreasing trends for SSR, SD and DTR.
Springtime (SON) presented statistically significant decreasing trends also for SD and
DTR. For SCF, statistically significant positive trends were observed for JJA and SON
only.

Table 2 - Modified Mann-Kendall trend test results for P     eriod 1, from 1961 to 1983,
and P     eriod 2, from 1984 to 2016, considering each season and in an annual basis for
the surface solar radiation (SSR), sunshine duration (SD), diurnal temperature range
(DTR) and sky cover fraction (SCF). The trend was estimated as the slope of the linear
fit between the variable of interest and year.

| SSR | | | | | | |
|---|---|---|---|---|---|---|
| | Period 1: 1961-1983 | | | Period 2: 1984-2016 | | |
| Time interval | Trend[a] | Z | p | Trend[a] | Z | P |
| Annual | -0.42 | -1.74 | 0.081 | **-0.41** | **-3.18** | **0.001** |
| DJF | -0.66 | -1.11 | 0.267 | **-0.54** | **-2.62** | **0.009** |
| MAM | **-0.78** | **-2.48** | **0.013** | -0.26 | -1.72 | 0.085 |
| JJA | **-0.48** | **-1.98** | **0.048** | **-0.18** | **-1.97** | **0.049** |
| SON | -0.25 | -0.96 | 0.335 | **-0.58** | **-2.46** | **0.014** |
| SD | | | | | | |
| | Period 1: 1961-1983 | | | Period 2: 1984-2016 | | |




| Time interval | Trend[b] | Z | p | Trend[b] | Z | P |
|---|---|---|---|---|---|---|
| Annual | **-0.37** | **-3.41** | **0.001** | **0.11** | **2.13** | **0.033** |
| DJF | -0.41 | -1.06 | 0.291 | -0.01 | -0.12 | 0.905 |
| MAM | **-0.53** | **-2.27** | **0.023** | 0.22 | 1.61 | 0.107 |
| JJA | **-0.54** | **-3.38** | **0.001** | **0.20** | **2.06** | **0.039** |
| SON | **-0.47** | **-2.31** | **0.021** | 0.03 | 0.20 | 0.840 |

| DTR | | | | | | |
|---|---|---|---|---|---|---|
| | **Period 1: 1961-1983** | | | **Period 2: 1984-2016** | | |
| Time interval | Trend[c] | Z | p | Trend[c] | Z | P |
| Annual | **-0.49** | **-3.33** | **0.001** | 0.16 | 1.84 | 0.065 |
| DJF | -0.32 | -1.61 | 0.107 | 0.15 | 1.72 | 0.085 |
| MAM | **-0.58** | **-2.54** | **0.011** | 0.16 | 1.53 | 0.125 |
| JJA | **-0.61** | **-2.91** | **0.004** | 0.14 | 1.38 | 0.171 |
| SON | **-0.58** | **-2.64** | **0.008** | 0.02 | 0.17 | 0.865 |

| SCF | | | | | | |
|---|---|---|---|---|---|---|
| | **Period 1: 1961-1983** | | | **Period 2: 1984-2016** | | |
| Time interval | Trend[d] | Z | p | Trend[d] | Z | P |
| Annual | **2.9** | **2.48** | **0.013** | 0.8 | 1.78 | 0.075 |
| DJF | 0.5 | 0.42 | 0.673 | 0.3 | 0.38 | 0.700 |
| MAM | 2.9 | 1.58 | 0.113 | 0.6 | 0.76 | 0.448 |
| JJA | **3.5** | **2.54** | **0.011** | 0.8 | 0.57 | 0.566 |
| SON | **3.8** | **2.12** | **0.034** | 1.5 | 1.22 | 0.221 |

Units of trend: a) kJ m$^{-2}$ per decade; b) hours per decade; c) °C per decade; d)
% per decade







In the first period, SSR and its proxies presented trends consistent with SFC
features, i.e., as SFC increased over time, the others decreased. In the second period,
from 1984 to 2016, this behavior combination changed. While SSR still presented, on
an annual basis, a statistically significant decreasing trend, of -0.41 kJm$^{-2}$ per decade,
SD and DTR trends changed from negative to positive, being statistically significant
only for SD, with a trend of 0.11 hours per decade. SFC continued to present a positive
trend, but not statistically significant. It is worth noting that, even though the trends are
not statistically significant, the pattern between SSR and SFC observed in the first
period remained in the second, and in all seasons. According to Rosas et al. (2019),
statistically significant trends, positive for low clouds (3.2% per decade) and negative
for mid level clouds (-5.5% per decade), were observed in the last 30 years, from 1987
to 2016. Such analysis indicated that changes in cloud types also influenced the
variability of SSR and proxies. However, other factors, rather than only cloud changes,
were also responsible for the variability of SD and DTR, as analyzed in the next
sections.
**3.2 Analysis of cloud free days**
From the good correlation between SSR and SFC, and based on previous results
from Yamasoe et al. (2017), cloud cover seems to be the main driver of SSR attenuation
in São Paulo. To evaluate the solely contribution of aerosol direct effect, we relied on a
limited number of completely clear sky days since the current study was based on
irradiation data, i.e., integrated from sunrise to sunset.  However, in order to have a clue
on its effect, mean atmospheric transmittance was estimated, during cloud free
conditions, i.e., considering only days with SCF less than 0.1 and with, at least, 9 cloud
free days per year. Most of those days were observed in winter and beginning of spring,
when dry conditions prevail, aerosol loading related to local sources is higher and when



biomass burning plumes from long range transport can be detected in São Paulo
(Castanho and Artaxo, 2001, Landulfo et al., 2003, Freitas et al., 2005, Castanho et al.,
2008, Yamasoe et al., 2017). For these reasons, we restricted this analysis using data
from July to October only.
For the first period, the cloud free mean transmittance was $0.691 \pm 0.029$ and for
the second period, a mean value of $0.700 \pm 0.023$ was estimated. Applying the Student
t-test to compare the two means, we obtained t = -0.87 and p = 0.40, thus, the null
hypothesis cannot be rejected at the 95% significance level, indicating that under
cloudless sky the mean atmospheric transmittance over São Paulo was similar in both
periods, suggesting that changes in the aerosol direct effect were unlikely to explain the
distinct features observed in both periods. Nevertheless, from Figure 3, which illustrates
the mean atmospheric transmittance (SSR/TSR) in cloud free conditions (i. e., SCF <=
0.1), in the first period, transmittance values were above 0.68, except in 1963, while in
the second period transmittance below 0.68 were more frequent, which might suggest
an increase in the atmospheric turbidity, particularly during the 1990's decade.
However, it is worth mention a recovery to higher transmittance values after 2010.
Similar features were also observed in n/N and horizontal visibility time series.

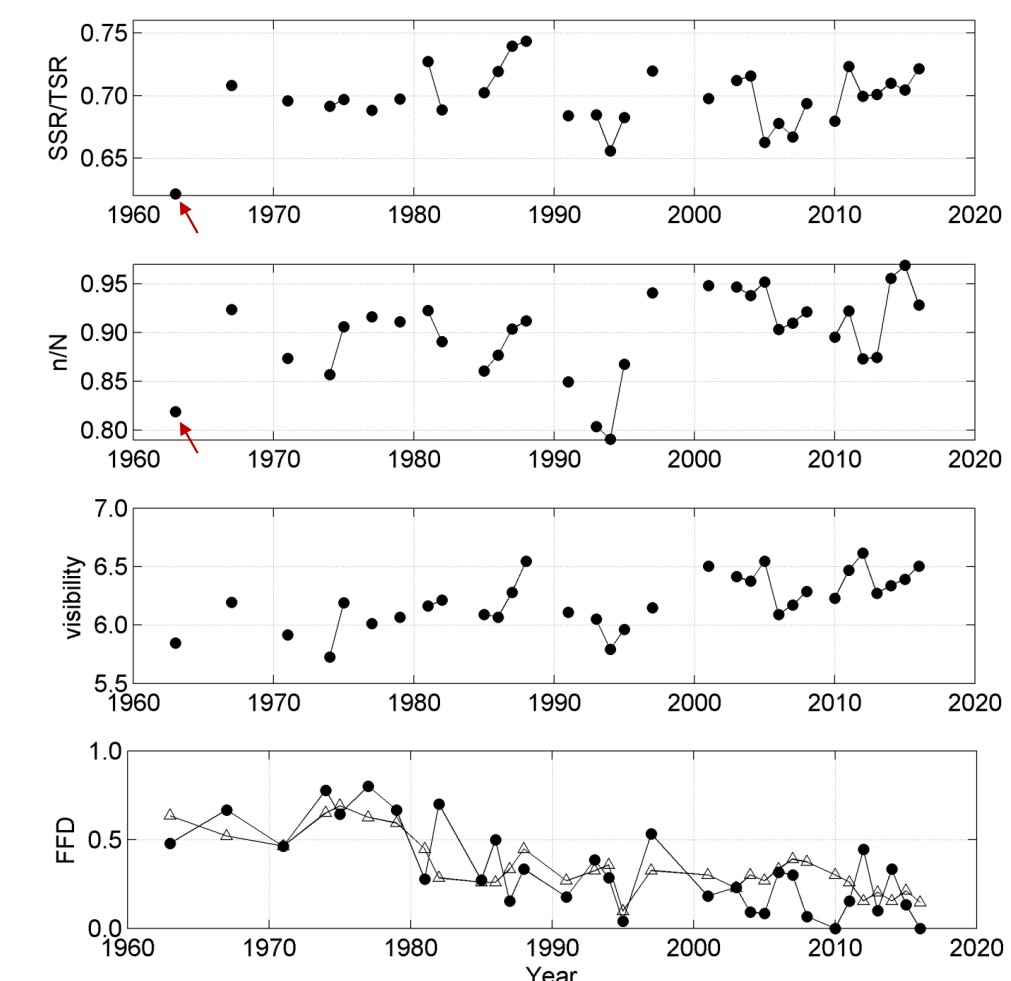



Figure 3 – Mean variability of cloud-free (SCF <= 0.1) atmospheric transmittance
(SSR/TSR), normalized sunshine duration (n/N), horizontal visibility and fraction of
foggy days (FFD) from July to September in each year (open symbols) and on the
cloud-free days only in the same period (full symbols). The red arrow indicates the year
of volcano Agung eruption, in 1963, whose signal was detected in both SSR and SD
data. Only years with more than 9 cloud-free days were considered.


Figure 3 also shows that 1963 presented the lowest mean transmittance in the

series, and a decrease observed in the normalized sunshine duration series as well.





According to Robertson et al. (2001), following Sato et al. (1993), one possible
explanation is the eruption of volcano Agung, whose plume affected southern latitudes,
with stratospheric AOD above 0.1 even one year after the eruption. Pinatubo eruption in
1991 also contributed to a high load of stratospheric AOD around latitude 25° S,
particularly one year after eruption, but no clear evidence was detected in our data.
Mean values of n/N varied from 0.841 ± 0.035, in the first period, to
0.852 ± 0.047, depicting a higher variability in the second one. Student t-test returned a
t value of -0.71, with p = 0.49, again indicating no difference in both periods.  In
contrast, horizontal visibility mean value varied from 6.04 ± 0.17 to 6.27 ± 0.21 and the
Student t-test returned t = -3.21 and p = 0.005, indicating that horizontal visibility in the
second period was statistically higher than in the first period, at the 95% significance
level. Both n/N and horizontal visibility for cloudless sky presented an increasing trend
particularly after 2000 (Figure 3). A possible explanation for this behavior may be due
to a reduction over time in the frequency of haze, fog and mist. Notice that
transmittance is more sensitive to haze than n/N, since haze can last throughout the day,
affecting continuously the transmittance, while, for the conditions observed in São
Paulo, its efficiency to extinguish the direct solar beam is limited, therefore, yielding a
lower impact on sunshine duration measurements. According to Stanhill et al. (2014),
only when aerosol optical depth (AOD) exceeds 2 sunshine duration recorders can be
sensitive to aerosol loadings and only early in the morning and late in the afternoon
(Horseman et al., 2008). By contrast, fog exerts a significant effect on n/N, because its
strongest impact occurs early in the morning when it is more frequent and when mostly
of solar radiation is in the diffuse component. Moreover, the number of days with fog is
decreasing in São Paulo, and particularly on the analyzed cloud free days, the fraction of
foggy days (FFD) decreased throughout the years as illustrated in Figure 3, what can



explain the increase of n/N in the recent years. This could also be the reason for the
positive trend of SD under all sky scenarios in the second period (Figure 2), when the
SFC increase was not significant. A decrease in the annual number of foggy days was
also observed in China (Li et al., 2012), which the authors attributed to the urban heat
island effect. As expected, horizontal visibility is also affected by the presence of fog,
although from Figure 3, only fog cannot explain all the variability observed in cloudless
sky conditions. During the late 1980's to early 1990's, transmittance, n/N and horizontal
visibility presented a significant decay clearly not related to the decrease observed in the
number of foggy days.
Concerning the urban heat island effect, the Metropolitan Area of São Paulo
experienced a fast growth rate from 1980 to 2010. There were nearly 12 million
inhabitants in 1980, and the population grew to about 21 million inhabitants in 2010
(Silva et al., 2017). According to the authors, the urban area increased from 874 km$^2$ to
2209 km$^2$, from 1962 to 2002. According to Kim and Baik (2002), the maximum UHI
intensity is more pronounced in clear sky conditions, occurs more frequently at night
than during the day, and decreases with increasing wind speed. However, Ferreira et al.
(2012) reported that, in São Paulo, the urban heat island maximum effect was observed
during day time, around 03:00 PM, and was associated with downward solar radiation
heating the urban region in a more effective way than the rural surrounding areas.
**3.3 Long term trends in daily maximum and minimum temperatures**
Figure 4 presents the temporal variation of the annual mean of the daily
maximum and minimum temperatures registered at the meteorological station, used to
estimate DTR. As discussed in the last paragraphs, if the increasing trend in SD over the
last years could be possibly attributed to the decreasing number of days per year with
fog occurrence, we now hypothesize on the possible reasons for the increasing trend of





DTR in the second period. According to Dai et al. (1999), it should also respond to
cloud cover and precipitation and thus to SSR variations. As discussed by the authors,
clouds can reduce $T_{max}$ and increase $T_{min}$, since they can reflect solar radiation back to
space and emit thermal radiation down to the surface, respectively. Such behaviors can
be clearly seen in Figure 4, in the first period, and confirmed by the trend analysis
presented in Table 3. During the dimming period, $T_{max}$ presented a negative trend, while
$T_{min}$ an increasing one, statistically significant at 95% confidence level for the last
variable. Similar behavior was observed by Wild et al. (2007) who argued that the
decreasing trend of $T_{max}$ is consistent with the negative trend of SSR, demonstrating that
solar radiation deficit at the surface presented a clear effect on the surface temperature.
Looking at the second period, from 1984 to 2016, both maximum and minimum
temperatures presented increasing trend, statistically significant at the 95% confidence
level, in the annual basis, of 0.25 °C per decade and 0.16 °C per decade, respectively. In
this period, $T_{min}$ trend was still in line with the increasing SFC trend, but as pointed out
by Wild et al. (2007) could also be a response to the increasing levels of greenhouse
gases as also pointed by de Abreu et al. (2019).

Table 3 - Modified Mann-Kendall trend test results for period 1, from 1961 to 1983, and
period 2, from 1984 to 2016, considering each season and in an annual basis, for the
daily maximum ($T_{max}$) and minimum ($T_{min}$) temperatures. The trend was estimated as
the slope of the linear fit between the variable of interest and year.

| | **$T_{max}$** | | | | | |
| | **Period 1: 1961-1983** | | | **Period 2: 1984-2016** | | |
| **Time interval** | **Trend** | **Z** | **p** | **Trend** | **Z** | **P** |
|---|---|---|---|---|---|---|
| **Annual** | -0.11 | -1.33 | 0.184 | **0.25** | **2.15** | **0.031** |
| **DJF** | 0.20 | 1.06 | 0.291 | **0.33** | **2.07** | **0.038** |





| | Trend | Z | p | Trend | Z | P |
|---|---|---|---|---|---|---|
| MAM | -0.15 | -0.79 | 0.430 | 0.03 | 0.23 | 0.816 |
| JJA | 0.02 | 0.26 | 0.795 | **0.33** | **2.68** | **0.007** |
| SON | -0.26 | -0.63 | 0.526 | 0.36 | 1.72 | 0.085 |

| | $T_{min}$ | | | | | |
|---|---|---|---|---|---|---|
| | **Period 1: 1961-1983** | | | **Period 2: 1984-2016** | | |
| **Time interval** | **Trend** | **Z** | **p** | **Trend** | **Z** | **P** |
| **Annual** | **0.56** | **2.54** | **0.011** | **0.16** | **2.15** | **0.031** |
| **DJF** | **0.53** | **2.96** | **0.003** | **0.13** | **2.68** | **0.007** |
| **MAM** | **0.52** | **2.71** | **0.007** | -0.07 | -0.79 | 0.429 |
| **JJA** | 0.62 | 1.58 | 0.113 | 0.26 | 1.78 | 0.075 |
| **SON** | -0.03 | 0.63 | 0.526 | **0.26** | **2.43** | **0.015** |

Units of trend:  °C per decade

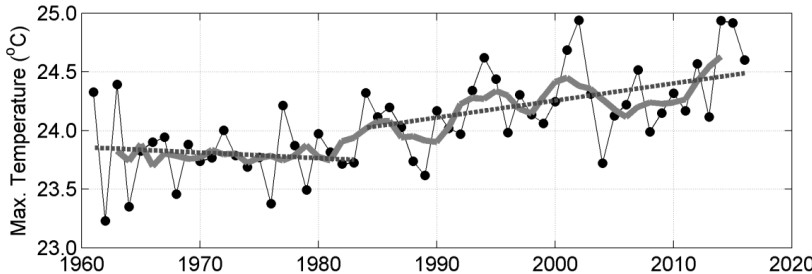

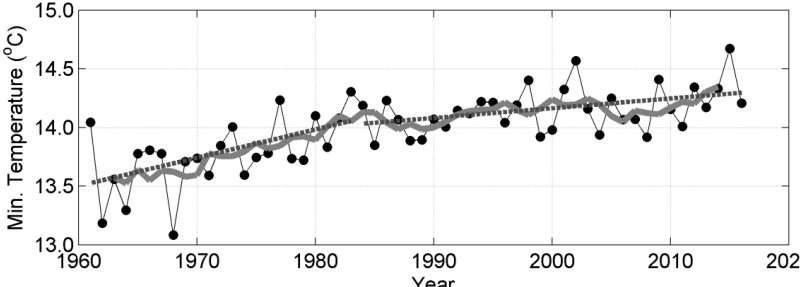


Figure 4 - Annual mean variability of daily maximum and minimum air temperatures at
1.5 meters. Gray curves represent 5 years moving averages and dotted lines are the
result of trend analysis from 1961 to 1983 and from 1984 to 2016.





From the previous discussion, although completely cloud free days were
extremely rare in São Paulo, the increase in $T_{max}$ in the second period can be attributed
to SSR changes associated with the aerosol direct effect only if the aerosol composition
changed from a more scattering to a more absorbing one, with a similar attenuation
effect on the solar radiation, as the atmospheric transmittance associated with aerosol
only was similar in both periods. A recent study by Andrade et al. (2017), discussing
changes over time in air quality conditions at the Metropolitan Area of São Paulo,
showed that $SO_2$ frequently exceeded the air quality standards in the 1980's. According
to the authors, the Brazilian government started a program to control its emission due to
the complaints of the population. At the beginning, the program focused on stationary
sources (industries) and, in the 1990's, the sulfur content in diesel fuel was also
targeted. Thus, as a consequence of this program, $SO_2$ concentrations declined and other
measures helped decreasing the concentration of particulate matter with diameter less
than 10 µm ($PM_{10}$) near the surface. However, according to Oyama (2015), also due to a
political decision to stimulate the economy, the annual number of registrations of new
gasoline fueled vehicles increased exponentially, jumping from about 3000 vehicles in
1988, peaking in 2000 with 150000 registrations, decreasing slowly after that, to about
60000 in 2012.
Changes in aerosol chemical composition and consequently optical properties,
from more scattering to more absorbing, without affecting the atmospheric
transmissivity on cloud free days could possibly explain the effect on $T_{max}$. Sulfate,
formed by gas to particle conversion of $SO_2$, is efficient as cloud condensation nuclei
(Easter and Hobbs, 1974) and also presents high single scattering albedo (Takemura et
al. 2002). Even with the renovation of the vehicular fleet in São Paulo, old heavy duty
vehicles fueled with diesel still circulate in the MASP area, and according to Andrade et



al. (2017) the diesel fleet constitute the main source of organic aerosols. In the case of
diesel fueled vehicles, the number of new registered vehicles in the São Paulo city
increased from about 5000 in 2000 to more than 25000 in 2010, the year with the
highest number of registrations (Oyama, 2015).  According to Feng et al. (2019),
toluene secondary organic aerosol (SOA) presents low single scattering albedo in the
ultraviolet-visible spectral range (0.78 ± 0.02) and toluene is one of the most abundant
among the aromatic volatile hydrocarbons present in gasoline and other fuels (Brocco et
al, 1997, Yamamoto et al., 2000). Particles with high absorption efficiency to solar
radiation, such as black carbon, can cause heating of the atmosphere. According to
Martins et al. (2009) aerosol particles measured during the wintertime of 1999 (August
and September) presented high absorption efficiency in the ultraviolet spectrum, even
higher than black carbon, which the authors attributed to the organic aerosol component.
Previous results, from the AERONET (Aerosol Robotic Network) radiometer operating
in the city, reported relative low single scattering albedo for aerosols from local sources,
SSA at 550 nm around 0.85, (Castanho et al., 2008, Yamasoe et al., 2017).
In order to verify the possibility of a pattern change in aerosol properties, from a
more scattering to a more absorbing one, without a significant change on aerosol
attenuation capacity, at least during the second period, annual mean values of absorbing
aerosol index and aerosol optical depth time series are presented in Figure 5. As
mentioned previously, data only for the months of July, August, September and October
were considered. For AAI, data are from 1979 to 2016 while for AOD, the MODIS in
2000 for Terra and 2002 for Aqua. Aerosol optical depth from MODIS onboard Terra
and Aqua satellites Figure 5 presents the annual mean values time series. From the
figure, annual mean AAI presented higher variability than mean AOD, particularly in
the 1980 and 1990 decades, varying from 0.1 to 0.6 in the period. AOD, by contrast,





varied from 0.13 to 0.28. Now, in order to verify possible trends, considering the second
period only, i.e., from 1984 to 2016, the modified Mann-Kendall trend test was applied.
A statistically significant positive trend of 0.07 AAI per decade, at 95% confidence
level, was observed (Z = 2.81 and p = 0.005), consistent with the discussion from the
previous paragraph. Since satellite retrieval of aerosol optical depth over land started
only during the 2000's, no trend analysis was applied.

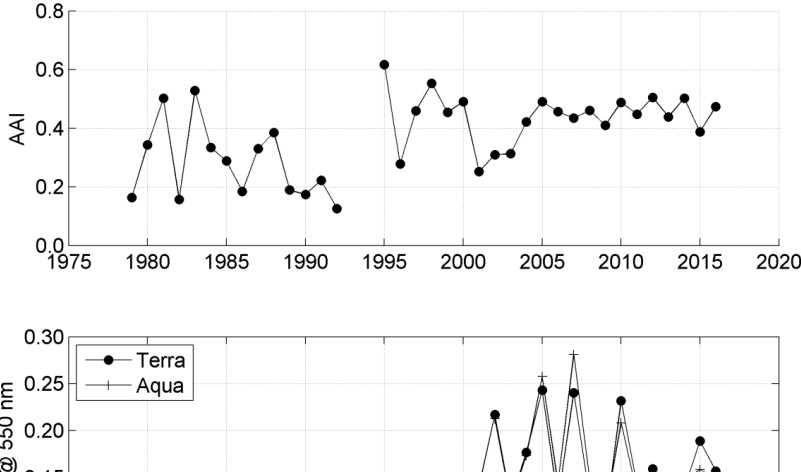


Figure 5 - Annual mean variability of absorbing aerosol index (AAI) (top) and aerosol
optical depth (AOD) from MODIS onboard Terra and Aqua satellites (bottom).

As discussed previously, due to the fast urbanization of the Metropolitan
Area of São Paulo (Silva et al., 2017), the urban heat island effect could also be
responsible to the  observed increasing trend of $T_{max}$, particularly after 1980. Finally, as
pointed by Wild et al. (2007), the increasing atmospheric concentration of greenhouse





gases (GHG) can be another reason for the observed trend of $T_{max}$, which was masked
by the dimming effect in the first period. Modeling studies can help verify the real
causes and disentangle the contribution of each effect, which is, however, out of the
scope of this work.

**4    Conclusions**
This analysis of 56 years of surface solar irradiation (SSR) and proxies (SD and
DTR) data helped to show that from about 1960 to early 1980, named as first period, a
dimming effect of surface solar radiation was observed in the city of São Paulo,
consistent to other parts of the world. The positive trend of SCF in the first period
indicates that cloud variability could be one important driver of the dimming period.
The dimming effect was also confirmed by SD and DTR trends in the mentioned
period. However, the consistency between SSR, SD and DTR trends ended in 1983,
when SCF presented the highest value throughout the entire series and which coincided
with a strong El Niño year. Thus, answering our first question, SSR presented a
decreasing trend, throughout the 56 years of data, though not statistically significant at
the 95% confidence level in the first period, while it decreased at a rate of -0.41 kJ m$^{-2}$
per decade in the second one, from 1984 to 2016.
In the second period, the negative SSR trend was still consistent with the slight
positive trend of SCF, while the opposite behavior of SD and DTR indicated that other
factors besides the cloud cover variability might have affected their distinct patterns. In
order to understand the possible causes of the SD trends, a restrict analysis of alternative
parameters (fog frequency and horizontal visibility) focusing on cloud free days, for the
dry months of July to October, were analyzed, in spite of the limited number of
available days per year even allowing some flexibility (SCF <= 0.1). The results



indicated that the decreasing trend of the number of foggy days per year is a potential
candidate to explain part of the increasing trend of SD and horizontal visibility.
Although on cloud free days, no statistically significant difference was observed
between SD in the first and the second period. Only horizontal visibility on cloud free
days presented a statistically significant increase from the first to the second period. The
analysis of cloud free days also showed that the effect of Agung volcano eruption was
detected in both SSR and SD annual mean values. Due to Agung eruption, in 1963, the
annual mean transmittance was the lowest in the series. In the case of DTR, since it was
obtained from the difference between the daily maximum and minimum air
temperatures close to the surface, the trends of the annual mean values of these
temperatures were separately determined and analyzed. The $T_{min}$ positive trends
followed the SCF ones, with also possible influence of the increasing levels of
greenhouse gases, noticing that the decay observed in SCF, in the beginning of the
second period, is absent in the $T_{min}$ time series. The increasing trend of SCF, in the first
period, resulted in a decreasing trend in $T_{max}$, as more solar radiation reaching the
surface was attenuated from year to year due to the presence of clouds. One hypothesis
for the increasing trend of $T_{max}$ during the second period was the changing of aerosol
optical properties in São Paulo, from a more scattering to a more absorbing one. Sulfate
particles, which scatter solar radiation with high efficiency, had the emission of
precursors to the atmosphere forced to decrease in the 1980's by governmental policies.
However, other political decisions, to promote economic development, caused the
increase of the gasoline fueled vehicles in São Paulo city in the beginning of the 1990's.
Gasoline and other fuels are important sources of toluene, whose SOA presents very
low single scattering albedo. The availability of an AERONET site in São Paulo, after
2000, made it possible to verify that the single scattering albedo of aerosol particles





from local sources can be quite low. Data of absorbing aerosol index retrieved from
multiple satellites since 1979 and aerosol optical depth from MODIS onboard Terra and
Aqua satellites were analyzed to verify the hypothesis of changing aerosol optical
properties. The modified Mann-Kendall trend analysis for the AAI showed that this
variable presented a positive trend statistically significant at 95% confidence level
during the second period, although no trend analysis for AOD was performed because
of the short time series available. Other hypotheses are the urban heat island effect and
the increasing concentrations of GHG. Of course, changes in the wind pattern and
consequently in the advection of air masses with distinct properties can also affect the
air temperature locally.
As the resultant trends of SD and DTR, compared with the SSR trend, diverged
in the second period for São Paulo, in all sky conditions, caution might be taken when
those variables are used as proxies to downward surface solar radiation in the context of
dimming and brightening analyses. This study revealed that different factors may act on
each variable, leading to a distinct behavior, as also mentioned by Manara et al. (2017).
For future studies, modeling efforts may be able to help evaluate each hypothesis
raised in the present study, either those related to climate natural variability, such as El
Niño, or to those arising from anthropogenic activities as the increase of greenhouse gas
concentrations, land use changes, particularly through the imperviousness of soils,
affecting the partitioning of latent and sensible heat fluxes. Also, higher temporal
analysis and simultaneous monitoring of aerosol optical properties will help to better
evaluate the aerosol effects on downward solar radiation in this region, including via the
indirect effect.

**Data availability**



Access to IAG meteorological station database (sky cover fraction, sunshine duration,
daily maximum and mimimum air temperatures, number of foggy days, visibility and
irradiation data) for education or scientific use can be made under request at
http://www.estacao.iag.usp.br/sol_dados.php. The multi-sensor absorbing aerosol index
was downloaded from http://www.temis.nl/airpollution/absaai/#MS_AAI, while AOD
from MODIS on board Terra and Aqua satellites were obtained from
https://giovanni.gsfc.nasa.gov/giovanni/. All processed data used in the manuscript such
as annual and seasonal mean values, as well as data from cloud free days can be found
at https://www.iag.usp.br/lraa/index.php/data/cientec/weather-station-climatology/.

**Author contribution**
Conceptualization MAY and NMER; Methodology MAY; Data organization MAY and
SNSMA; Formal analysis MAY; Writing original draft MAY; Writing – Review &
Editing MAY, NMER, MW.

**Competing interest**
The authors declare that they have no conflict of interest.

***Acknowledgements***
The authors acknowledge Fundação de Amparo à Pesquisa do Estado de São Paulo
(FAPESP), grant number 2018/16048-6 and Coordenação de Aperfeiçoamento de
Pessoal de Nível Superior (CAPES) for financial support. Yamasoe acknowledges
CNPq (Conselho Nacional de Desenvolvimento Cientifico e Tecnologico), process



number 313005/2018-4. This study is part of the Núcleo de Apoio à Pesquisa em
Mudanças Climáticas (INCLINE). The authors are grateful to the observers and staff of
the Instituto de Astronomia, Geofísica e Ciências Atmosféricas meteorological station
for making available the meteorological observations.

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
