# Peer review of "Fifty-six years of Surface Solar Radiation and Sunshine"

_Atmospheric Chemistry and Physics, 2020_

## Referee Comment (RC1) · Anonymous Referee #2 · 25 Oct 2020

The paper by Akemi Yamasoe et al. presents the results of the analysis of a 56-year record of surface downward solar irradiation with respect to other atmospheric parameters for São Paulo, Brazil. The authors try to define the main drivers of changes in irradiation during the period of study. Although some of the findings are interesting, improvement is necessary prior to publication. More specific comments are provided below.

L50: Since the two different trends are not a global phenomenon (e.g. even some of the referred studies show different results for China and India), I suggest adding "over wide regions of the world" or something similar after "documented".

[Figure]

L58: Zerefos et al. (2009) could be also cited at this point (in addition to Wild 2012):

ZEREFOS, C.S., ELEFTHERATOS, K., MELETI, C., KAZADZIS, S., ROMANOU, A., ICHOKU, C., TSELIOUDIS, G. and BAIS, A. (2009), Solar dimming and brightening over Thessaloniki, Greece, and Beijing, China. Tellus B, 61: 657-665. doi:10.1111/j.1600-0889.2009.00425.x

L65-66: Relative discussion (regarding the main drivers of the trends over particular areas) can be also found in:

- Kazadzis, S., Founda, D., Psiloglou, B. E., Kambezidis, H., Mihalopoulos, N., Sanchez-Lorenzo, A., Meleti, C., Raptis, P. I., Pierros, F., and Nabat, P.: Long-term series and trends in surface solar radiation in Athens, Greece, Atmos. Chem. Phys., 18, 2395–2411, https://doi.org/10.5194/acp-18-2395-2018, 2018.

- Manara, V., Brunetti, M., Celozzi, A., Maugeri, M., Sanchez-Lorenzo, A., and Wild, M.: Detection of dimming/brightening in Italy from homogenized all-sky and clear-sky surface solar radiation records and underlying causes (1959–2013), Atmos. Chem. Phys., 16, 11145–11161, https://doi.org/10.5194/acp-16-11145-2016, 2016.

- Manara, V., Bassi, M., Brunetti, M. et al. 1990–2016 surface solar radiation variability and trend over the Piedmont region (northwest Italy). Theor Appl Climatol 136, 849–862 (2019). https://doi.org/10.1007/s00704-018-2521-6

L92: Delete "Thus,"

L118: Please define if this is the standard (k=1) or the expanded (k=2) uncertainty.

L135 – 147: I am very skeptical about the methodology used to study the effect of aerosols. The authors have used a very small number of cloud-free days for each year in the period July – October in order to study the effect of aerosol. I doubt that with such a small number of days (i.e. 9 days for some years) the authors can get safe conclusions. Furthermore, I do not think that the results can be generalized for the whole year.

L172: The AOD from MODIS at which wavelength?

L173 – 175: "Shortly . . . spectrum". Please add the appropriate reference.

L177 – 178: Again, I believe that the authors should analyze and discuss the AOD and the AAI for different seasons in the year, and for the whole year. This way they would also provide some evidence for what they claim, i.e. that the effect of aerosol is significant only in July – October.

L168 – 183: Some discussion regarding the uncertainties in the AOD and AAI product would be useful.

L211:" it is listed amongst the 24 strongest El Niño events". The 24 strongest events during which time period?

L227 – 229: "After 1983, the trend behavior of all variables changed". What does this phrase mean? How do the authors define the "change"? While it is acceptable to study the trends for two different periods, the authors cannot support that there is a change in the trends without any further statistical analysis. What I mean is that someone could argue that e.g. the trend in SSR did not change at all, or that the trend in SD changed in 1980. If the authors want to support their statement that "the trend changed" in a particular year, or period of years, they should use more robust statistical analysis. See for example the methodology used by Yang et al., 2006 in order to investigate whether there is a statistically significant change in the trends of stratospheric ozone: Yang, E.S., Cunnold, D. M., Salawitch, R. J., McCormick, M. P., Russell, J., Zawodny, J. M., Oltmans, S., and Newchurch, M. J. (2006), Attribution of recovery in lower‐stratospheric ozone, J. Geophys. Res., 111, D17309, doi:10.1029/2005JD006371.

L242 and L243: "Period" instead of "P eriod"

Section 3.2: Again, my main concern regarding the analysis for the effect of clouds is that the number of cloudless days is too small. So, the results may be misleading. I don't know if making the analysis for a particular time in the day (e.g. local noon?)

could give more reliable results.

Section 3.3 Since AOD and SSA measurements from AERONET are available at Sao Paolo since 2000, I suggest that they should be also used in the analysis. The AOD measurements could be even used to evaluate the MODIS product. The AERONET data will give more information relative to the fact that: "In the case of diesel fueled vehicles, the number of new registered vehicles in the São Paulo city increased from about 5000 in 2000 to more than 25000 in 2010, the year with the highest number of registrations"

L385 – 390: As the number of cloud-free days analyzed for each year is small, I do not think that the authors can be sure that AOD did not change.

---

## Referee Comment (RC2) · Anonymous Referee #1 · 26 Oct 2020

General comments

The paper discusses the long-term trends of downwelling solar irradiance at earth's surface in Sao Paolo, Brazil, one of the longest periods of such observations world-wide (1964-2016). The authors have identified in this dataset the well-known from other studies global dimming period of surface irradiance up the end of the 1980s. However, the brightening period that has been found in other locations over the world has not been confirmed at this station. The authors using other ancillary information, such as sunshine duration, cloudiness, diurnal temperature range, and days of fog have attempted to explain, to a certain degree, this unexpected behavior. The scien-

tific questions addressed in this manuscript are well within the scopes of ACP. The innovation of the paper lies in the uniqueness of the dataset as well as in the use of measurements of additional geophysical parameters in order to test and explain their findings.

The methods of data analysis are quite standard for this type of studies but could be improved, as suggested in the specific comments below, especially as far as it concerns the use of deseasonalized data to derive the annual means for the calculation of the trends. Overall, the paper is well structured and presented with adequate clarity, although there is room for further improvements. Most of the conclusions drawn from the results are supported by appropriate references.

Generally, the language of the paper is adequate, but some parts should be be further improved to enhance readability. I have provided suggestions for some cases in the Technical Comments section, but there are more sentences that need fixing. Particular effort should be put to the Conclusions section which seems to have been written hastily with and several sentences are difficult to read.

Specific comments

Title: I suggest rephrasing to:

Fifty-six years of Surface Solar Radiation and Sunshine Duration over São Paulo, Brazil: 1961–2016

or

Long term changes of Surface Solar Radiation and Sunshine Duration over São Paulo, Brazil (1961–2016)

Line 17: Please include in the abstract some quantitative estimates of the trends in the two periods, at least for solar irradiation. The abstract is quite generic expressing mainly the intentions and not so much the findings.

[Figure]

47: The acronym SSR is defined here as surface solar radiation while later in line 88 is defined as surface solar irradiation. Please fix this because it is important to have a clear distinction between the two quantities.

89: The term "cloud cover fraction" (CCF) is more common in literature, and in essence, than term "sky cover fraction"

118: Has the calibration of the instrument been monitored during the 56 years of operation? From the cited reference (1988) I understand that the 5% uncertainty characterizes the type of this instrument and does not include the uncertainty of the long-term stability of the instrument's sensitivity. Please discuss this in more detail.

126: Annual averages are biased by the high summer values therefore are not representative for the year. I suggest using monthly anomalies (deviations from the long term monthly mean) and from them to calculate the annual means and derive the trends. This approach will probably alter the significance level of the trends.

146: It is not clear whether the 9-day limit refers to each month (July to October) or to the entire 4-month period.

148: Please clarify whether in the calculation of the atmospheric transmittance the solar irradiance (TSR) been adjusted for the variation of sun-earth distance.

153: As the station is located about 800 m above sea level, I assume that in many cases fog may occur below this altitude and on these occasions it would not affect the solar radiation measured and the station. Are these conditions distinguishable in the dataset?

154-155: Please clarify whether days with fog have been excluded from the clear sky averages.

155: Is the "fraction of cloud free days with foggy conditions" the FFD used in figure 3? If not, please explain how this index has been calculated. Figure 3 suggests that FFD can be as high as 0.8. Would this mean that in the particular year 80% of clear days

are foggy?

159: Table 1 could be removed because it does not add any information that is used in the analysis.

187: Are the annual averages of the different variables computed for the common days of data or for each variable all available data have been included? This might influence the results in case of a large number of missing observations.

196: In Figure 3 the upward trend in cloud cover does not extend to 1988 and ends in 1983. Is 1988 a typographical error or there is really a difference between the total cloud cover (this paper) and the trend of the two cloud types reported in Rosas 2019?

274-275: Please clarify whether the threshold of 0.1 for the cloud fraction refers to the average of all measurement during the day or to each measurement during the day.

274-275: Please state how the limit of 9 cloud free days per year has been determined. Isn't it too small, representing only $\sim$2.5% of the available days? Is it related to the 2nd percentile representing the absolute maximum of the data?

280: July-October: Fig 3 caption states July-September. Which of the two is correct?

297: Figure 3: It would be interesting to show how the DTR is behaving for clear-sky conditions.

306: Visibility in 1963 is also quite low (possibly related to the Agung eruption?), which may have partly contributed to the reduction of SSR in this year.

325: I don't understand what is meant by "the AOD exceeds 2 sunshine duration recorders". Please rephrase.

328-329: I cannot understand why effects on sunshine duration will be stronger when most of radiation is in the diffuse component. I would expect the opposite, i.e. that under prevalence of diffuse radiation the sunshine recorder would be less sensitive and effects of fog would not make any difference.

341-350: This section discusses the heat island effect which is not relevant to trends in cloud free irradiation. It would better fit in the next section where it could be connected to temperature changes and DTR or in the introduction.

370: The increasing trend in the daily minimum temperature is indeed qualitatively in line with increasing cloudiness, but the latter is very small and insignificant (Table 2). The heat island discussed for the fog trend should have also played a role in the temperature trend.

381: Figure 4: It would be interesting to show how these variables behave for clear skies only. A second set of lines with clear-sky values could be added with different color or symbol.

385-390: Please try to split this long sentence in to two. It is difficult to read.

428: Please state the wavelength of the aerosol optical depth data.

442: Does the SSA from AERONET show any trend after 2000? What about the AOD from this instrument?

433: The variability of the AAI and the AOD cannot be compared in absolute terms because these two qualities are not the same.

436: From Figure 5 is evident that there is an abrupt change in AAI between 1992 and 1994 which should not be neglected. The AAI after 1995 has been almost doubled and remained fairly constant. Considering the years 1984-2016 as one period for a trend is probably not a good choice since the trend is not linear.

470: "their distinct patterns". If I understand correctly, the other factors may have affected the SSR and not the SD and DTR, thus it should be changed to "the distinct changes in SSR".

503-506: See my comment for line 436 above.

Technical

30: Delete "still"

34: Replace "encouraged" by "planned"

55: Insert "comprising" before "both", and delete the two occurrences of "in"

67: Replace "the" by "increasing"

92: Replace "propose to answer" with "are addressing"

95-97: Better use "section" instead of "part"

148: Insert "by" after "estimated"

150: Delete "also"

182: Please define that the dry season is July-October.

242-243: There is a long blank after the P in word Period (two occurences)

270: Replace "solely" with "sole"

271: Replace "clue" by "quantitative estimate"

275: Please add after "spring" the months corresponding to winter and spring season, just to avoid confusion for the readers living in the northern hemisphere.

291: Delete "decade"

292: Replace "mention" by "mentioning"

293: Replace "n/N" by normalized sunshine duration"

331: Delete (FFD) as it is has been already defined

333: Please replace "scenarios" with "conditions"

339: Replace "decay" with "reduction" since decay usually implies a gradual decrease but here we have a rather abrupt change.

357: Replace "it" by "DTR"

360: Please add after "space", "during daytime" and after "surface", "during the night"

400: Delete "jumping"

401: Add "and" before "decreasing"

423: Replace "relative" with "relatively"

431,432: Something is missing in this sentence.

434: Please replace "1980 and 1990 decades" with "in the 1980s and 1990s".

471: Delete "a restrict analysis of"

475-476: Delete "is a potential candidate to"

477: Replace "Although" with "However,"

---

## Author Comment (AC1) · 15 Feb 2021

We would like to thank both anonymous referees for the devoted time evaluating and reviewing our manuscript, with constructive suggestions to improve the final version. Each comment is addressed as follows, with the answer provided bellow:

Referee #1 The paper by Akemi Yamasoe et al. presents the results of the analysis of a 56-year record of surface downward solar irradiation with respect to other atmospheric parameters for São Paulo, Brazil. The authors try to define the main drivers of changes in irradiation during the period of study. Although some of the findings are interesting, improvement is necessary prior to publication. More specific comments are provided

[Figure]

below.

1. L50: Since the two different trends are not a global phenomenon (e.g. even some of the referred studies show different results for China and India), I suggest adding "over wide regions of the world" or something similar after "documented".

Authors response: The text was added as suggested.

2. L58: Zerefos et al. (2009) could be also cited at this point (in addition to Wild 2012): ZEREFOS, C.S., ELEFTHERATOS, K., MELETI, C., KAZADZIS, S., RO-MANOU, A., ICHOKU, C., TSELIOUDIS, G. and BAIS, A. (2009), Solar dimming and brightening over Thessaloniki, Greece, and Beijing, China. Tellus B, 61: 657-665. doi:10.1111/j.1600-0889.2009.00425.x

Authors response: Yes, indeed the suggested reference complements the discussion about the geographical heterogeneity of the "brightening phase" comparing UV-A and total solar irradiances at Thessaloniki, in Greece, and Beijing, in China.

3. L65-66: Relative discussion (regarding the main drivers of the trends over particular areas) can be also found in: - Kazadzis, S., Founda, D., Psiloglou, B. E., Kambezidis, H., Mihalopoulos, N., Sanchez-Lorenzo, A., Meleti, C., Raptis, P. I., Pierros, F., and Nabat, P.: Long-term series and trends in surface solar radiation in Athens, Greece, Atmos. Chem. Phys., 18, 2395–2411, https://doi.org/10.5194/acp-18-2395-2018, 2018. - Manara, V., Brunetti, M., Celozzi, A., Maugeri, M., Sanchez-Lorenzo, A., and Wild, M.: Detection of dimming/brightening in Italy from homogenized all-sky and clear-sky surface solar radiation records and underlying causes (1959–2013), Atmos. Chem. Phys., 16, 11145–11161, https://doi.org/10.5194/acp-16-11145-2016, 2016. - Manara, V., Bassi, M., Brunetti, M. et al. 1990–2016 surface solar radiation variability and trend over the Piedmont region (northwest Italy). Theor Appl Climatol 136, 849– 862 (2019). https://doi.org/10.1007/s00704-018-2521-6

Authors response: We appreciated your suggestions and added the one below besides

the suggested references: Yang, S., Wang, X. L. and Wild, M. Causes of Dimming and Brightening in China Inferred from Homogenized Daily Clear-Sky and All-Sky in situ Surface Solar Radiation Records (1958-2016). Journal of Climate 32, 5901-5913, doi: 10.1175/JCLI-D-18-0666.1, 2019.

4. L92: Delete "Thus,"

Authors response: Word deleted.

5. L118: Please define if this is the standard (k=1) or the expanded (k=2) uncertainty.

Authors response: It is the standard (k = 1) instrumental uncertainty. More information concerning SSR uncertainty and long-term shift of the actinograph calibration, please, see reply to item 22 for referee #2 below.

6. L135 – 147: I am very skeptical about the methodology used to study the effect of aerosols. The authors have used a very small number of cloud-free days for each year in the period July – October in order to study the effect of aerosol. I doubt that with such a small number of days (i.e. 9 days for some years) the authors can get safe conclusions. Furthermore, I do not think that the results can be generalized for the whole year.

Authors response: We agreed with both referees that the number of cloud-free days for each year is not enough for a robust statistical analysis. For this reason, we changed this part of the manuscript, also modifying the discussion on the aerosol effect. Instead, we replaced with a discussion on visibility, using it as a proxy for aerosol optical depth and the number of foggy days during the same period, i. e. from July to October. In this time of the year, the aerosol can have a stronger effect on SSR, due to reduced cloud fraction, higher aerosol loadings either because of more stable conditions and less precipitation allowing air pollution to build up or due to long range transport of smoke from vegetation fires in other parts of South America. Occurrence of fog is also more frequent.

7. L172: The AOD from MODIS at which wavelength?

Authors response: The AOD analysis was removed. The aerosol impact on SSR deserves a careful analysis and with a more appropriate database.

8. L173 – 175: "Shortly : : : spectrum". Please add the appropriate reference.

Authors response: This part of the text was removed.

9. L177 – 178: Again, I believe that the authors should analyze and discuss the AOD and the AAI for different seasons in the year, and for the whole year. This way they would also provide some evidence for what they claim, i.e. that the effect of aerosol is significant only in July – October.

Authors response: This part of the text was removed.

10. L168 – 183: Some discussion regarding the uncertainties in the AOD and AAI product would be useful.

Authors response: This part of the text was removed.

11. L211:" it is listed amongst the 24 strongest El Niño events". The 24 strongest events during which time period?

Authors response: According to the Earth System Research Laboratory from the National Oceanic and Atmospheric Administration (ESRL/NOAA) the time period is from 1895 to 2015 (https://www.esrl.noaa.gov/psd/enso/climaterisks/years/top24enso.html). This period was included in the manuscript to make it clear.

12. L227 – 229: "After 1983, the trend behavior of all variables changed". What does this phrase mean? How do the authors define the "change"? While it is acceptable to study the trends for two different periods, the authors cannot support that there is a change in the trends without any further statistical analysis. What I mean is that someone could argue that e.g. the trend in SSR did not change at all, or that the trend in SD changed in 1980. If the authors want to support their statement that "the

trend changed" in a particular year, or period of years, they should use more robust statistical analysis. See for example the methodology used by Yang et al., 2006 in order to investigate whether there is a statistically significant change in the trends of stratospheric ozone: Yang, E.S., Cunnold, D. M., Salawitch, R. J., McCormick, M. P., Russell, J., Zawodny, J. M., Oltmans, S., and Newchurch, M. J. (2006), Attribution of recovery in lowerâĚŸARĚĞ stratospheric ozone, J. Geophys. Res., 111, D17309, doi:10.1029/2005JD006371.

Authors response: We agree with the referee comment, modifying and complementing the statement to (see further discussion below): "After 1983, the trend behaviour of some variables changed, consistent with the findings of Reid et al. (2016), who observed a regime shift in land surface temperature anomalies in South America in 1984".

Applying different statical analyses, we verified that Tmax presented a statistically significant ($p = 8.5 \times 10^{-7}$) regime shift in 1984, according to the method proposed by Rodionov (2004), based on mean values. Using the package "segmented" from R (Muggeo, 2003), only SD and DTR presented a shift, in 1982 ($p = 0.008$) and in 1979 ($p = 0.017$), respectively, indicating that depending on the variable and the methodology, the trend change can be detected in different years. The observed change in Tmax, in 1984, is consistent with the findings of Reid et al. (2016). The authors evaluated 72 time series around the world to analyse the 1980s regime shift. They observed that the shift was first observed in South America, in 1984, and spread toward North Pacific and North America (1985), to the North Atlantic Ocean (1986), Europe (1987) and Asia (1988). In the Southern Hemisphere, it extended eastwards to the Indian Ocean in 1986 and Australia in 1987. Shortly, one hypothesis is that it was a combination of factors, from a recovery of the cooling effect caused by El Chichón volcano eruption in 1982, with a natural warming, which intensified the anthropogenic warming due to greenhouse gas emissions.

The cited references are:

Muggeo, V. M. R. Estimating regression models with unknown break-points. Statist. Med. 22, 3055–3071. doi: 10.1002/sim.1545, 2003.

Reid, P. C., Hari, R. E., Beaugrand, G., Livingstone, D. M., Marty, C., Straile, D., Barichivich, J., Goberville, E., Adrian, R., Aono, Yasuyuki, Brown, R., Foster, J. Groisman, P., Hélaouët, P., Hsu, H.-H., Kirby, R., Knight, J., Kraberg, A., Li, J., Lo, T.-T., Myneni, R. B., North, R. P., Pounds, J. A., Sparks, T., Stübi, R., Tian, Y., Wiltshire, K. H., Xiao, D. and Zhu, Z. Global impacts of the 1980s regime shift. Global Change Biology 22, 682-703, doi: 10.1111/gcb.13106, 2016.

Rodionov, S. N. A sequential algorithm for testing climate regime shifts. Geophysical Research Letters 31, L09204. doi: 10.1029/2004GL019448, 2004.

13. L242 and L243: "Period" instead of "P eriod"

Authors response: Corrected.

14. Section 3.2: Again, my main concern regarding the analysis for the effect of clouds is that the number of cloudless days is too small. So, the results may be misleading. I don't know if making the analysis for a particular time in the day (e.g. local noon?) could give more reliable results.

Authors response: Section 3.2 was modified, excluding the analysis of cloud free days. Now, in that section we discuss the long-term trend of visibility, as a proxy for aerosol optical depth and the number of foggy days. To separate each effect, we used visibility data recorded from 10:00 AM to 03:00 PM, as fog is more frequent early in the morning and can impact visibility.

15. Section 3.3 Since AOD and SSA measurements from AERONET are available at Sao Paolo since 2000, I suggest that they should be also used in the analysis. The AOD measurements could be even used to evaluate the MODIS product. The AERONET data will give more information relative to the fact that: "In the case of diesel fueled vehicles, the number of new registered vehicles in the São Paulo city increased from

about 5000 in 2000 to more than 25000 in 2010, the year with the highest number of registrations"

Authors response: As explained previously, this part of the manuscript was removed.

16. L385 – 390: As the number of cloud-free days analyzed for each year is small, I do not think that the authors can be sure that AOD did not change.

Authors response: We agree with the referee and for this reason, we removed the AOD analysis.

Referee # 2: 17. General comments The paper discusses the long-term trends of downwelling solar irradiance at earth's surface in Sao Paolo, Brazil, one of the longest periods of such observations worldwide (1964-2016). The authors have identified in this dataset the well-known from other studies global dimming period of surface irradiance up the end of the 1980s. However, the brightening period that has been found in other locations over the world has not been confirmed at this station. The authors using other ancillary information, such as sunshine duration, cloudiness, diurnal temperature range, and days of fog have attempted to explain, to a certain degree, this unexpected behavior. The scientific questions addressed in this manuscript are well within the scopes of ACP. The innovation of the paper lies in the uniqueness of the dataset as well as in the use of measurements of additional geophysical parameters in order to test and explain their findings. The methods of data analysis are quite standard for this type of studies but could be improved, as suggested in the specific comments below, especially as far as it concerns the use of deseasonalized data to derive the annual means for the calculation of the trends. Overall, the paper is well structured and presented with adequate clarity, although there is room for further improvements. Most of the conclusions drawn from the results are supported by appropriate references. Generally, the language of the paper is adequate, but some parts should be be further improved to enhance readability. I have provided suggestions for some cases in the Technical Comments section, but there are more sentences that need fixing. Particular effort should be put to the Conclusions section which seems to have been written hastily with and several sentences are difficult to read.

Authors response: We appreciated and thank referee #2 for the comments and suggestions. For the deseasonalized analysis, we made some tests as explained below (item 23) and observed only negligible differences. The manuscript was revised and we believe that the readability was improved in the revised version. Thank you for calling our attention to this.

Specific comments 18. Title: I suggest rephrasing to: Fifty-six years of Surface Solar Radiation and Sunshine Duration over São Paulo, Brazil: 1961–2016 or Long term changes of Surface Solar Radiation and Sunshine Duration over São Paulo, Brazil (1961–2016)

Authors response: The first suggestion was accepted, and the title was changed accordingly.

19. Line 17: Please include in the abstract some quantitative estimates of the trends in the two periods, at least for solar irradiation. The abstract is quite generic expressing mainly the intentions and not so much the findings.

Authors response: As suggested, we included quantitative estimates of the trends for solar irradiation and cloud cover in lines 28, 29, 31 and 32.

20. 47: The acronym SSR is defined here as surface solar radiation while later in line 88 is defined as surface solar irradiation. Please fix this because it is important to have a clear distinction between the two quantities.

Authors response: We removed the acronym SSR from line 47 and used it only when referring to surface solar irradiation, which is the variable analysed in the manuscript.

21. 89: The term "cloud cover fraction" (CCF) is more common in literature, and in essence, than term "sky cover fraction"

[Figure]

Authors response: We changed the term, as suggested.

22. 118: Has the calibration of the instrument been monitored during the 56 years of operation? From the cited reference (1988) I understand that the 5% uncertainty characterizes the type of this instrument and does not include the uncertainty of the long-term stability of the instrument's sensitivity. Please discuss this in more detail.

Authors response: Yes, the 5% refers to the instrumental uncertainty. To verify the stability of the instrument, in 2014, a one-year comparison experiment was conducted with a brand new Robitzsch-Fuess Actinograph, type 58dc. Monthly scatterplots comparing irradiation data from daily measurements are available at http://www.estacao.iag.usp.br/Relatorios/Relat_tecnico_3.pdf (in Portuguese). Figure 1 presents the scatterplot of all daily measurements performed in 2014, comparing both instruments data. Bars (vertical and horizontal) indicate the 5% instrumental uncertainty, as pointed out by the referee. Applying a least square fitting to the data, the resulting slope is $0.919 \pm 0.006$. Assuming that the operational actinograph suffered a linear degradation throughout the years, from 1961 to 2014, and that, in the first year, the slope should be equal 1, the difference in slope resulted in a long-term trend of about -1.5 % per decade, ranging from (-1.6 to -1.4) % per decade. This agrees with the result by Plana-Fattori and Ceballos (1988) indicating that the hypothesis of a linear degradation can be considered. In order to take this long-term shift in the instrument calibration, we applied this linear trend to SSR data. This changed the results as presented in the Table below, particularly in JJA, for which the trends are now outside the 5% significance level. Table 2 (Table 1 in the revised version) of the manuscript was updated taking this correction into account as well as SSR in the Figure 2. We added this discussion as supplementary information to the manuscript.

Figure 1 – Comparison of one year of irradiation measurements performed with two Actinographs, the operational, whose data is discussed in the manuscript, and a brand new one. The blue line represents the least square fit with coefficients: linear = $0.30 \pm 0,07$ and slope = $0.919 \pm 0.006$. The red line is the 1:1.

Table 1 - Modified Mann-Kendall trend test results for Period 1, from 1961 to 1983, and Period 2, from 1984 to 2016, considering each season and in an annual basis for the surface solar radiation (SSR) in units of kJ m^-2 per decade. Period 1: 1961-1983 Period 2: 1984-2016 Time interval Trend Z p Trend Z p Annual -0.40 -1.64 0.101 -0.39 -3.02 0.003 DJF -0.64 -1.05 0.291 -0.53 -2.56 0.010 MAM -0.76 -2.48 0.013 -0.25 -1.66 0.097 JJA -0.47 -1.93 0.054 -0.17 -1.87 0.061 SON -0.24 -0.89 0.373 -0.57 -2.40 0.016

23. 126: Annual averages are biased by the high summer values therefore are not representative for the year. I suggest using monthly anomalies (deviations from the long term monthly mean) and from them to calculate the annual means and derive the trends. This approach will probably alter the significance level of the trends.

Authors response: Only negligible difference was observed at the significance level of the trends if using the monthly anomalies. Please, see the example for SSR annual trends at Table 2 and compare with the results presented at Table 1. We attributed this to the low number of missing values. For SSR, the total number of missing days was 59 (from a total of 20454 days), the maximum number of missing days was five per year and four per month. Sunshine duration, maximum and minimum air temperatures presented no missing data and only two days in the entire series was missing for the cloud cover fraction.

Table 2 – Modified Mann-Kendall trend results using monthly anomalies data for SSR Period 1: 1961-1983 Period 2: 1984-2016 Time interval Trend Z p Trend Z p Annual -0.40 -1.69 0.091 -0.39 -2.99 0.003

24. 146: It is not clear whether the 9-day limit refers to each month (July to October) or to the entire 4-month period.

Authors response: It referred to the entire 4-month period. For this reason, we agreed with both referees that it was too low for a robust statistical analysis and decided to remove the clear sky analysis in the revised version of the manuscript.

[Figure]

25. 148: Please clarify whether in the calculation of the atmospheric transmittance the solar irradiance (TSR) been adjusted for the variation of sun-earth distance.

Authors response: Yes, and we adopted the empirical formulas proposed by Paltridge and Platt (1976) to take that into account. We added this information in the manuscript to make it clear.

26. 153: As the station is located about 800 m above sea level, I assume that in many cases fog may occur below this altitude and on these occasions it would not affect the solar radiation measured and the station. Are these conditions distinguishable in the dataset?

Authors response: When fog is reported at the meteorological station, it is observed at surface level. Although the site is located at 800 m above sea level, fog can form due to radiative cooling at night. In effect, the mean elevation of São Paulo Metropolitan Area is around 800 m due to its location on a plateau, the Brazilian Plateau.

27. 154-155: Please clarify whether days with fog have been excluded from the clear sky averages.

Authors response: The clear sky analysis was removed from the revised version of the manuscript.

28. 155: Is the "fraction of cloud free days with foggy conditions" the FFD used in figure 3? If not, please explain how this index has been calculated. Figure 3 suggests that FFD can be as high as 0.8. Would this mean that in the particular year 80% of clear days are foggy?

Authors response: Yes. It means that in the particular year, from July to October, 80% of clear days are foggy. But as pointed by both referees, the low number of clear sky days (from 9 to 23) can result in this kind of artificial result, motivating us to remove this analysis from the final version of the manuscript.

29. 159: Table 1 could be removed because it does not add any information that is

used in the analysis.

Authors response: We removed the table as suggested.

30. 187: Are the annual averages of the different variables computed for the common days of data or for each variable all available data have been included? This might influence the results in case of a large number of missing observations.

Authors response: As mentioned previously, for SSR, the total number of missing days was 59 (from a total of 20454 days), the maximum number of missing days was five per year and four per month. Sunshine duration, maximum and minimum air temperatures presented no missing data and only two days in the entire series was missing for the cloud cover fraction.

31. 196: In Figure 3 the upward trend in cloud cover does not extend to 1988 and ends in 1983. Is 1988 a typographical error or there is really a difference between the total cloud cover (this paper) and the trend of the two cloud types reported in Rosas 2019?

Authors response: We believe that the referee meant in Figure 2. Rosas et al. (2019) conducted the cloud cover database analysis considering visual observations from 1958 to 2016, and they evaluated the trends for the first 29 (from 1958 to 1986) and the last 30 years (from 1987 to 2016), and also for the whole time series. The increasing trend for stratiform cloud fraction of 4.8 % per decade and of 1.4 % per decade, in the case of cirrus clouds, were observed in the period from 1958 to 1986. Thus, 1988 was a typographical error. As explained in comment 12 to referee # 1, we separated the series in the 1983-1984 for the reasons discussed in the manuscript and which coincided to the trend shift detected in South America by Reid et al. (2016).

32. 274-275: Please clarify whether the threshold of 0.1 for the cloud fraction refers to the average of all measurement during the day or to each measurement during the day.

Authors response: It refers to each measurement during the day.

33. 274-275: Please state how the limit of 9 cloud free days per year has been determined. Isn't it too small, representing only 2.5% of the available days? Is it related to the 2nd percentile representing the absolute maximum of the data?

Authors response: Yes, it is indeed too small. Cloud free days are rare in São Paulo, particularly in the afternoon. For this reason, we removed the analysis of cloud free days from the manuscript.

34. 280: July-October: Fig 3 caption states July-September. Which of the two is correct?

Authors response: The correct is July-October.

35. 297: Figure 3: It would be interesting to show how the DTR is behaving for clear-sky conditions.

Authors response: Due to the low number of data, we excluded the analysis for clear sky conditions.

36. 306: Visibility in 1963 is also quite low (possibly related to the Agung eruption?), which may have partly contributed to the reduction of SSR in this year.

Authors response: Yes, we believe that the low visibility in 1963 also contributed to the reduction of SSR in that year, but we are not sure if it is also related to the Agung eruption.

37. 325: I don't understand what is meant by "the AOD exceeds 2 sunshine duration recorders". Please rephrase.

Authors response: This part of the manuscript was removed.

38. 328-329: I cannot understand why effects on sunshine duration will be stronger when most of radiation is in the diffuse component. I would expect the opposite, i.e. that under prevalence of diffuse radiation the sunshine recorder would be less sensitive and effects of fog would not make any difference.

Authors response: This part of the manuscript was removed.

39. 341-350: This section discusses the heat island effect which is not relevant to trends in cloud free irradiation. It would better fit in the next section where it could be connected to temperature changes and DTR or in the introduction.

Authors response: The discussion was moved to the next section, in the context of temperature changes.

40. 370: The increasing trend in the daily minimum temperature is indeed qualitatively in line with increasing cloudiness, but the latter is very small and insignificant (Table 2). The heat island discussed for the fog trend should have also played a role in the temperature trend.

Authors response: As mentioned in the previous comment, we discussed the urban heat island effect in the context of temperature trends.

41. 381: Figure 4: It would be interesting to show how these variables behave for clear skies only. A second set of lines with clear-sky values could be added with different color or symbol.

Authors response: As the number of clear sky days was too low for a statistically significant analysis, it was removed from the manuscript as pointed by both referees.

42. 385-390: Please try to split this long sentence in to two. It is difficult to read.

Authors response: We removed this part of the discussion in the revised version of the manuscript. The effect of aerosol on SSR will be analysed in a future study, when more data related to aerosol properties could be gathered.

43. 428: Please state the wavelength of the aerosol optical depth data.

Authors response: Text removed from the revised manuscript.

44. 442: Does the SSA from AERONET show any trend after 2000? What about the

AOD from this instrument?

Authors response: Text removed from the revised manuscript.

45. 433: The variability of the AAI and the AOD cannot be compared in absolute terms because these two qualities are not the same.

Authors response: Text removed from the revised manuscript.

46. 436: From Figure 5 is evident that there is an abrupt change in AAI between 1992 and 1994 which should not be neglected. The AAI after 1995 has been almost doubled and remained fairly constant. Considering the years 1984-2016 as one period for a trend is probably not a good choice since the trend is not linear.

Authors response: Text removed from the revised manuscript.

47. 470: "their distinct patterns". If I understand correctly, the other factors may have affected the SSR and not the SD and DTR, thus it should be changed to "the distinct changes in SSR".

Authors response: At this point we meant that other factors may have affected SSR (such as aerosol optical depth, analysed by the variability of visibility data), SD (number of foggy days) and DTR (urban heat island effect and anthropogenic greenhouse gas emissions).

48. 503-506: See my comment for line 436 above.

Authors response: Removed from the final version.

49. Technical 30: Delete "still" - OK 34: Replace "encouraged" by "planned" - OK 55: Insert "comprising" before "both", and delete the two occurrences of "in" - OK 67: Replace "the" by "increasing" - OK 92: Replace "propose to answer" with "are addressing" - OK 95-97: Better use "section" instead of "part" - OK 148: Insert "by" after "estimated" - OK 150: Delete "also" - OK 182: Please define that the dry season is July-October. As previously discussed, the analysis of the aerosol effect was removed

from the manuscript. 242-243: There is a long blank after the P in word Period (two occurences) - OK 270: Replace "solely" with "sole" – this sentence was removed from the revised version of the manuscript. 271: Replace "clue" by "quantitative estimate" – the text was removed from the revised version of the manuscript. 275: Please add after "spring" the months corresponding to winter and spring season, just to avoid confusion for the readers living in the northern hemisphere. – the text was removed from the revised version of the manuscript. 291: Delete "decade" – the text was removed from the revised version of the manuscript. 292: Replace "mention" by "mentioning" – the text was removed from the revised version of the manuscript. 293: Replace "n/N" by normalized sunshine duration" – the text was removed from the revised version of the manuscript. 331: Delete (FFD) as it is has been already defined – the text was removed from the revised version of the manuscript. 333: Please replace "scenarios" with "conditions" – the text was removed from the revised version of the manuscript. 339: Replace "decay" with "reduction" since decay usually implies a gradual decrease but here we have a rather abrupt change. – the text was removed from the revised version of the manuscript. 357: Replace "it" by "DTR" - OK 360: Please add after "space", "during daytime" and after "surface", "during the night" - OK 400: Delete "jumping" - OK 401: Add "and" before "decreasing" - OK 423: Replace "relative" with "relatively" – the text was removed from the revised version of the manuscript. 431,432: Something is missing in this sentence. – the text was removed from the revised version of the manuscript. 434: Please replace "1980 and 1990 decades" with "in the 1980s and 1990s". – the text was removed from the revised version of the manuscript. 471: Delete "a restrict analysis of" – OK. 475-476: Delete "is a potential candidate to" - OK 477: Replace "Although" with "However," – the text was removed from the revised version of the manuscript.

Please also note the supplement to this comment:
https://acp.copernicus.org/preprints/acp-2020-848/acp-2020-848-AC1-supplement.pdf

[Figure]

[Figure]

[Figure]

**Fig. 1.**

**Supplement:**

**Supplementary Information**

**Verification of the long-term calibration shift of the Actinograph**

To evaluate the stability of the Actinograph used to measure surface solar irradiation (SSR) analysed in the manuscript, we compare a one-year of simultaneous measurements, conducted in 2014, with a brand new Robitzsch-Fuess Actinograph, type 58dc. Monthly scatterplots comparing irradiation data from daily measurements are available at http://www.estacao.iag.usp.br/Relatorios/Relat_tecnico_3.pdf (in Portuguese – last access on 10 February, 2021). Figure S.1 presents the scatterplot of all daily measurements performed in 2014, comparing both instruments data, where SSR_operational refers to irradiation measured by the instrument whose data is analysed in the manuscript and SSR_new refers to irradiation measured by the new actinograph. Bars (vertical and horizontal) indicate the 5% instrumental uncertainty. Applying a least square fitting to the data, the resulting slope was 0.919 ± 0.006. We assume that the operational actinograph suffered a linear degradation throughout the years, from 1961 to 2014, and that, in the first year, the slope should be equal 1. Using such hypotheses, the difference in slope results in a long-term trend of about -1.5 % per decade, ranging from (-1.6 to -1.4) % per decade. Identical results were obtained by Plana-Fattori and Ceballos (1988), confirming the long-term stability of the instrument.

[Figure]

Figure S.1 – Comparison of one year of irradiation measurements performed with two Actinographs, the operational, whose data is discussed in the manuscript, and a brand

new one. The blue line represents the least square fit with coefficients: linear = 0.30 ± 0,07 and slope = 0.919 ± 0.006. The red line is the 1:1.

---

## Author Response (AR1)

We would like to thank both anonymous referees for the devoted time evaluating and reviewing our manuscript, with constructive suggestions to improve the final version. Each comment is addressed as follows, with the answer provided in bold characters:

**Referee #1**

The paper by Akemi Yamasoe et al. presents the results of the analysis of a 56-year record of surface downward solar irradiation with respect to other atmospheric parameters for São Paulo, Brazil. The authors try to define the main drivers of changes in irradiation during the period of study. Although some of the findings are interesting, improvement is necessary prior to publication. More specific comments are provided below.

1. L50: Since the two different trends are not a global phenomenon (e.g. even some of the referred studies show different results for China and India), I suggest adding "over wide regions of the world" or something similar after "documented".

**A: The text was added as suggested. See line 53 of the revised version of the manuscript.**

2. L58: Zerefos et al. (2009) could be also cited at this point (in addition to Wild 2012):

ZEREFOS, C.S., ELEFTHERATOS, K., MELETI, C., KAZADZIS, S., ROMANOU, A., ICHOKU, C., TSELIOUDIS, G. and BAIS, A. (2009), Solar dimming and brightening over Thessaloniki, Greece, and Beijing, China. Tellus B, 61: 657-665. doi:10.1111/j.1600-0889.2009.00425.x

**A: Yes, indeed the suggested reference complements the discussion about the geographical heterogeneity of the "brightening phase" comparing UV-A and total solar irradiances at Thessaloniki, in Greece, and Beijing, in China.**

**See line 62.**

3. L65-66: Relative discussion (regarding the main drivers of the trends over particular areas) can be also found in:

- Kazadzis, S., Founda, D., Psiloglou, B. E., Kambezidis, H., Mihalopoulos, N., Sanchez-Lorenzo, A., Meleti, C., Raptis, P. I., Pierros, F., and Nabat, P.: Long-term series and trends in surface solar radiation in Athens, Greece, Atmos. Chem. Phys., 18, 2395–2411, https://doi.org/10.5194/acp-18-2395-2018, 2018.

- Manara, V., Brunetti, M., Celozzi, A., Maugeri, M., Sanchez-Lorenzo, A., and Wild, M.: Detection of dimming/brightening in Italy from homogenized all-sky and clear-sky surface solar radiation records and underlying causes (1959–2013), Atmos. Chem. Phys., 16, 11145–11161, https://doi.org/10.5194/acp-16-11145-2016, 2016.

- Manara, V., Bassi, M., Brunetti, M. et al. 1990–2016 surface solar radiation variability and trend over the Piedmont region (northwest Italy). Theor Appl Climatol 136, 849– 862 (2019). https://doi.org/10.1007/s00704-018-2521-6

**A: We appreciated your suggestions and added the one below besides the suggested references:**

**Yang, S., Wang, X. L. and Wild, M. Causes of Dimming and Brightening in China Inferred from Homogenized Daily Clear-Sky and All-Sky in situ Surface Solar Radiation Records (1958-2016). Journal of Climate 32, 5901-5913, doi: 10.1175/JCLI-D-18-0666.1, 2019.**

**See lines 70 and 71.**

4. L92: Delete "Thus,"

**A: Word deleted. See line 98 of the revised version.**

5. L118: Please define if this is the standard (k=1) or the expanded (k=2) uncertainty.

**A: It is the standard (k = 1) instrumental uncertainty. More information concerning SSR uncertainty and long-term shift of the actinograph calibration, please, see reply to item 22 for referee #2 below.**

**See lines 124 to 129 of the revised version of the manuscript.**

6. L135 – 147: I am very skeptical about the methodology used to study the effect of aerosols. The authors have used a very small number of cloud-free days for each year in the period July – October in order to study the effect of aerosol. I doubt that with such a small number of days (i.e. 9 days for some years) the authors can get safe conclusions. Furthermore, I do not think that the results can be generalized for the whole year.

**A: We agreed with both referees that the number of cloud-free days for each year is not enough for a robust statistical analysis. For this reason, we changed this part of the manuscript, also modifying the discussion on the aerosol effect. Instead, we replaced with a discussion on visibility, using it as a proxy for aerosol optical depth and the number of foggy days during the same period, i. e. from July to October. In this time of the year, the aerosol can have a stronger effect on SSR, due to reduced cloud fraction, higher aerosol loadings either because of more stable conditions and less precipitation allowing air pollution to build up or due to long range transport of smoke from vegetation fires in other parts of South America. Occurrence of fog is also more frequent.**

**The whole section 3.2 was modified (lines 261 to 348).**

7. L172: The AOD from MODIS at which wavelength?

**A: The AOD analysis was removed. The aerosol impact on SSR deserves a careful analysis and with a more appropriate database.**

8. L173 – 175: "Shortly : : : spectrum". Please add the appropriate reference.

**A: This part of the text was removed.**

9. L177 – 178: Again, I believe that the authors should analyze and discuss the AOD and the AAI for different seasons in the year, and for the whole year. This way they would also provide some evidence for what they claim, i.e. that the effect of aerosol is significant only in July – October.

**A: This part of the text was removed.**

10. L168 – 183: Some discussion regarding the uncertainties in the AOD and AAI product would be useful.

**A: This part of the text was removed.**

11. L211:" it is listed amongst the 24 strongest El Niño events". The 24 strongest events during which time period?

**A: According to the Earth System Research Laboratory from the National Oceanic and Atmospheric Administration (ESRL/NOAA) the time period is from 1895 to 2015 (https://www.esrl.noaa.gov/psd/enso/climaterisks/years/top24enso.html). This period was included in the manuscript to make it clear. See lines 206 and 207 of the revised version of the manuscript.**

12. L227 – 229: "After 1983, the trend behavior of all variables changed". What does this phrase mean? How do the authors define the "change"? While it is acceptable to study the trends for two different periods, the authors cannot support that there is a change in the trends without any further statistical analysis. What I mean is that someone could argue that e.g. the trend in SSR did not change at all, or that the trend in SD changed in 1980. If the authors want to support their statement that "the trend changed" in a particular year, or period of years, they should use more robust statistical analysis. See for example the methodology used by Yang et al., 2006 in order to investigate whether there is a statistically significant change in the trends of stratospheric ozone: Yang, E.S., Cunnold, D. M., Salawitch, R. J., McCormick, M. P., Russell, J., Zawodny, J. M., Oltmans, S., and

Newchurch, M. J. (2006), Attribution of recovery in lowerâ˘ARˇ stratospheric ozone, J. Geophys. Res., 111, D17309, doi:10.1029/2005JD006371.

**A: We agree with the referee comment, modifying and complementing the statement to (see further discussion below): "After 1983, the trend behaviour of some variables changed, consistent with the findings of Reid et al. (2016), who observed a regime shift in land surface temperature anomalies in South America in 1984". This part of the text can be found at lines 222 to 224 of the revised version of the manuscript.**

**Applying different statical analyses, we verified that Tmax presented a statistically significant ($p = 8.5 \times 10^{-7}$) regime shift in 1984, according to the method proposed by Rodionov (2004), based on mean values. Using the package "segmented" from R (Muggeo, 2003), only SD and DTR presented a shift, in 1982 ($p = 0.008$) and in 1979 ($p = 0.017$), respectively, indicating that depending on the variable and the methodology, the trend change can be detected in different years. The observed change in Tmax, in 1984, is consistent with the findings of Reid et al. (2016). The authors evaluated 72 time series around the world to analyse the 1980s regime shift. They observed that the shift was first observed in South America, in 1984, and spread toward North Pacific and North America (1985), to the North Atlantic Ocean (1986), Europe (1987) and Asia (1988). In the Southern Hemisphere, it extended eastwards to the Indian Ocean in 1986 and Australia in 1987. Shortly, one hypothesis is that it was a combination of factors, from a recovery of the cooling effect caused by El Chichón volcano eruption in 1982, with a natural warming, which intensified the anthropogenic warming due to greenhouse gas emissions.**

**The cited references are:**

**Muggeo, V. M. R. Estimating regression models with unknown break-points. Statist. Med. 22, 3055–3071. doi: 10.1002/sim.1545, 2003.**

**Reid, P. C., Hari, R. E., Beaugrand, G., Livingstone, D. M., Marty, C., Straile, D., Barichivich, J., Goberville, E., Adrian, R., Aono, Yasuyuki, Brown, R., Foster, J. Groisman, P., Hélaouët, P., Hsu, H.-H., Kirby, R., Knight, J., Kraberg, A., Li, J., Lo, T.-T., Myneni, R. B., North, R. P., Pounds, J. A., Sparks, T., Stübi, R., Tian, Y.,**

**Wiltshire, K. H., Xiao, D. and Zhu, Z. Global impacts of the 1980s regime shift. Global Change Biology 22, 682-703, doi: 10.1111/gcb.13106, 2016.**

**Rodionov, S. N. A sequential algorithm for testing climate regime shifts. Geophysical Research Letters 31, L09204. doi: 10.1029/2004GL019448, 2004.**

13. L242 and L243: "Period" instead of "P eriod"

**A: It was corrected. See lines 238 and 239.**

14. Section 3.2: Again, my main concern regarding the analysis for the effect of clouds is that the number of cloudless days is too small. So, the results may be misleading. I don't know if making the analysis for a particular time in the day (e.g. local noon?) could give more reliable results.

**A: Section 3.2 was modified, excluding the analysis of cloud free days. Now, in that section we discuss the long-term trend of visibility, as a proxy for aerosol optical depth and the number of foggy days. To separate each effect, we used visibility data recorded from 10:00 AM to 03:00 PM, as fog is more frequent early in the morning and can impact visibility.**

15. Section 3.3 Since AOD and SSA measurements from AERONET are available at Sao Paolo since 2000, I suggest that they should be also used in the analysis. The AOD measurements could be even used to evaluate the MODIS product. The AERONET data will give more information relative to the fact that: "In the case of diesel fueled vehicles, the number of new registered vehicles in the São Paulo city increased from about 5000 in 2000 to more than 25000 in 2010, the year with the highest number of registrations"

**A: As explained previously, this part of the manuscript was removed.**

16. L385 – 390: As the number of cloud-free days analyzed for each year is small, I do not think that the authors can be sure that AOD did not change.

**A: We agree with the referee and for this reason, we removed the AOD analysis.**

**Referee # 2:**

17. General comments

The paper discusses the long-term trends of downwelling solar irradiance at earth's surface in Sao Paolo, Brazil, one of the longest periods of such observations worldwide (1964-2016). The authors have identified in this dataset the well-known from other studies global dimming period of surface irradiance up the end of the 1980s. However, the brightening period that has been found in other locations over the world has not been confirmed at this station. The authors using other ancillary information, such as sunshine duration, cloudiness, diurnal temperature range, and days of fog have attempted to explain, to a certain degree, this unexpected behavior. The scientific questions addressed in this manuscript are well within the scopes of ACP. The innovation of the paper lies in the uniqueness of the dataset as well as in the use of measurements of additional geophysical parameters in order to test and explain their findings.

The methods of data analysis are quite standard for this type of studies but could be improved, as suggested in the specific comments below, especially as far as it concerns the use of deseasonalized data to derive the annual means for the calculation of the trends. Overall, the paper is well structured and presented with adequate clarity, although there is room for further improvements. Most of the conclusions drawn from the results are supported by appropriate references. Generally, the language of the paper is adequate, but some parts should be be further improved to enhance readability. I have provided suggestions for some cases in the Technical Comments section, but there are more sentences that need fixing. Particular effort should be put to the Conclusions section which seems to have been written hastily with and several sentences are difficult to read.

**A: We appreciated and thank referee #2 for the comments and suggestions. For the deseasonalized analysis, we made some tests as explained below (item 23) and**

**observed only negligible differences. The manuscript was revised and we believe that the readability was improved in the revised version. Thank you for calling our attention to this.**

Specific comments

18. Title: I suggest rephrasing to:

Fifty-six years of Surface Solar Radiation and Sunshine Duration over São Paulo,

Brazil: 1961–2016 or Long term changes of Surface Solar Radiation and Sunshine Duration over São Paulo, Brazil (1961–2016)

**A: The first suggestion was accepted, and the title was changed accordingly.**

19. Line 17: Please include in the abstract some quantitative estimates of the trends in the two periods, at least for solar irradiation. The abstract is quite generic expressing mainly the intentions and not so much the findings.

**A: As suggested, we included quantitative estimates of the trends for solar irradiation and cloud cover in lines 24, 25, 28, 29, 31, 32 and 33 of the revised version of the manuscript.**

20. 47: The acronym SSR is defined here as surface solar radiation while later in line 88 is defined as surface solar irradiation. Please fix this because it is important to have a clear distinction between the two quantities.

**A: We removed the acronym SSR from line 47 and used it only when referring to surface solar irradiation, which is the variable analysed in the manuscript. See line 50.**

21. 89: The term "cloud cover fraction" (CCF) is more common in literature, and in essence, than term "sky cover fraction"

**A: We changed the term, as suggested. See line 95 of the revised version of the manuscript.**

22. 118: Has the calibration of the instrument been monitored during the 56 years of operation? From the cited reference (1988) I understand that the 5% uncertainty characterizes the type of this instrument and does not include the uncertainty of the long-term stability of the instrument's sensitivity. Please discuss this in more detail.

**A: Yes, the 5% refers to the instrumental uncertainty. To verify the stability of the instrument, in 2014, a one-year comparison experiment was conducted with a brand new Robitzsch-Fuess Actinograph, type 58dc. Monthly scatterplots comparing irradiation data from daily measurements are available at http://www.estacao.iag.usp.br/Relatorios/Relat_tecnico_3.pdf (in Portuguese). Figure 1 presents the scatterplot of all daily measurements performed in 2014, comparing both instruments data. Bars (vertical and horizontal) indicate the 5% instrumental uncertainty, as pointed out by the referee. Applying a least square fitting to the data, the resulting slope is 0.919 ± 0.006. Assuming that the operational actinograph suffered a linear degradation throughout the years, from 1961 to 2014, and that, in the first year, the slope should be equal 1, the difference in slope resulted in a long-term trend of about -1.5 % per decade, ranging from (-1.6 to -1.4) % per decade. This agrees with the result by Plana-Fattori and Ceballos (1988) indicating that the hypothesis of a linear degradation can be considered. In order to take this long-term shift in the instrument calibration, we applied this linear trend to SSR data. This changed the results as presented in the Table below, particularly in JJA, for which the trends are now outside the 5% significance level. Table 2 (Table 1 in the revised version, line 238) of the manuscript was updated taking this correction into account as well as SSR in the Figure 2 (line 193 of the revised version of the manuscript). We added this discussion as supplementary information to the manuscript.**

[Figure]

**Figure 1 – Comparison of one year of irradiation measurements performed with two Actinographs, the operational, whose data is discussed in the manuscript, and a brand new one. The blue line represents the least square fit with coefficients: linear = 0.30 ± 0,07 and slope = 0.919 ± 0.006. The red line is the 1:1.**

**Table 1 - Modified Mann-Kendall trend test results for Period 1, from 1961 to 1983, and Period 2, from 1984 to 2016, considering each season and in an annual basis for the surface solar radiation (SSR) in units of kJ m$^{-2}$ per decade.**

| | Period 1: 1961-1983 | | | Period 2: 1984-2016 | | |
|---|---|---|---|---|---|---|
| **Time interval** | **Trend** | **Z** | **p** | **Trend** | **Z** | **p** |
| **Annual** | -0.40 | -1.64 | 0.101 | **-0.39** | **-3.02** | **0.003** |
| **DJF** | -0.64 | -1.05 | 0.291 | **-0.53** | **-2.56** | **0.010** |
| **MAM** | **-0.76** | **-2.48** | **0.013** | -0.25 | -1.66 | 0.097 |
| **JJA** | -0.47 | -1.93 | 0.054 | -0.17 | -1.87 | 0.061 |
| **SON** | -0.24 | -0.89 | 0.373 | **-0.57** | **-2.40** | **0.016** |

23. 126: Annual averages are biased by the high summer values therefore are not representative for the year. I suggest using monthly anomalies (deviations from the long term monthly mean) and from them to calculate the annual means and derive the trends. This approach will probably alter the significance level of the trends.

**A: Only negligible difference was observed at the significance level of the trends if using the monthly anomalies. Please, see the example for SSR annual trends at Table 2 and compare with the results presented at Table 1. We attributed this to the low number of missing values. For SSR, the total number of missing days was 59 (from a total of 20454 days), the maximum number of missing days was five per year and four per month. Sunshine duration, maximum and minimum air temperatures presented no missing data and only two days in the entire series was missing for the cloud cover fraction.**

**Table 2 – Modified Mann-Kendall trend results using monthly anomalies data for SSR**

| Time interval | Period 1: 1961-1983 | | | Period 2: 1984-2016 | | |
|---|---|---|---|---|---|---|
| | Trend | Z | p | Trend | Z | p |
| **Annual** | -0.40 | -1.69 | 0.091 | **-0.39** | **-2.99** | **0.003** |

24. 146: It is not clear whether the 9-day limit refers to each month (July to October) or to the entire 4-month period.

**A: It referred to the entire 4-month period. For this reason, we agreed with both referees that it was too low for a robust statistical analysis and decided to remove the clear sky analysis in the revised version of the manuscript.**

25. 148: Please clarify whether in the calculation of the atmospheric transmittance the solar irradiance (TSR) been adjusted for the variation of sun-earth distance.

**A: Yes, and we adopted the empirical formulas proposed by Paltridge and Platt (1976) to take that into account. We added this information in the manuscript to make it clear.**

**See lines 176 and 177.**

26. 153: As the station is located about 800 m above sea level, I assume that in many cases fog may occur below this altitude and on these occasions it would not affect the solar radiation measured and the station. Are these conditions distinguishable in the dataset?

**A: When fog is reported at the meteorological station, it is observed at surface level. Although the site is located at 800 m above sea level, fog can form due to radiative cooling at night. In effect, the mean elevation of São Paulo Metropolitan Area is around 800 m due to its location on a plateau, the Brazilian Plateau.**

27. 154-155: Please clarify whether days with fog have been excluded from the clear sky averages.

**A: The clear sky analysis was removed from the revised version of the manuscript.**

28. 155: Is the "fraction of cloud free days with foggy conditions" the FFD used in figure 3? If not, please explain how this index has been calculated. Figure 3 suggests that FFD can be as high as 0.8. Would this mean that in the particular year 80% of clear days are foggy?

**A: Yes. It means that in the particular year, from July to October, 80% of clear days are foggy. But as pointed by both referees, the low number of clear sky days (from**

**9 to 23) can result in this kind of artificial result, motivating us to remove this analysis from the final version of the manuscript.**

29. 159: Table 1 could be removed because it does not add any information that is used in the analysis.

**A: We removed the table as suggested.**

30. 187: Are the annual averages of the different variables computed for the common days of data or for each variable all available data have been included? This might influence the results in case of a large number of missing observations.

**A: As mentioned previously, for SSR, the total number of missing days was 59 (from a total of 20454 days), the maximum number of missing days was five per year and four per month. Sunshine duration, maximum and minimum air temperatures presented no missing data and only two days in the entire series was missing for the cloud cover fraction.**

31. 196: In Figure 3 the upward trend in cloud cover does not extend to 1988 and ends in 1983. Is 1988 a typographical error or there is really a difference between the total cloud cover (this paper) and the trend of the two cloud types reported in Rosas 2019?

**A: We believe that the referee meant in Figure 2. Rosas et al. (2019) conducted the cloud cover database analysis considering visual observations from 1958 to 2016, and they evaluated the trends for the first 29 (from 1958 to 1986) and the last 30 years (from 1987 to 2016), and also for the whole time series. The increasing trend for stratiform cloud fraction of 4.8 % per decade and of 1.4 % per decade, in the case of cirrus clouds, were observed in the period from 1958 to 1986. Thus, 1988 was a typographical error. As explained in comment 12 to referee # 1, we separated the series in the 1983-1984 for the reasons discussed in the manuscript and which coincided to the trend shift detected in South America by Reid et al. (2016).**

**See line 190 and lines 222 to 224 of the revised version of the manuscript.**

32. 274-275: Please clarify whether the threshold of 0.1 for the cloud fraction refers to the average of all measurement during the day or to each measurement during the day.

**A: It refers to each measurement during the day.**

33. 274-275: Please state how the limit of 9 cloud free days per year has been determined. Isn't it too small, representing only 2.5% of the available days? Is it related to the 2nd percentile representing the absolute maximum of the data?

**A: Yes, it is indeed too small. Cloud free days are rare in São Paulo, particularly in the afternoon. For this reason, we removed the analysis of cloud free days from the manuscript.**

34. 280: July-October: Fig 3 caption states July-September. Which of the two is correct?

**A: The correct is July-October. Now, instead of clear sky days only, visibility and number of fog days time series refer to all sky days, but still for the months of July to October and is presented in Figure 4, lines 328 to 331.**

35. 297: Figure 3: It would be interesting to show how the DTR is behaving for clear-sky conditions.

**A: Due to the low number of data, we excluded the analysis for clear sky conditions.**

36. 306: Visibility in 1963 is also quite low (possibly related to the Agung eruption?), which may have partly contributed to the reduction of SSR in this year.

**A: Yes, we believe that the low visibility in 1963 also contributed to the reduction of SSR in that year, but we are not sure if it is also related to the Agung eruption.**

37. 325: I don't understand what is meant by "the AOD exceeds 2 sunshine duration recorders". Please rephrase.

**A: This part of the manuscript was removed.**

38. 328-329: I cannot understand why effects on sunshine duration will be stronger when most of radiation is in the diffuse component. I would expect the opposite, i.e. that under prevalence of diffuse radiation the sunshine recorder would be less sensitive and effects of fog would not make any difference.

**A: This part of the manuscript was removed.**

39. 341-350: This section discusses the heat island effect which is not relevant to trends in cloud free irradiation. It would better fit in the next section where it could be connected to temperature changes and DTR or in the introduction.

**A: The discussion was moved to the next section, in the context of temperature changes.**

**See lines 386 to 395 of the revised version of the manuscript.**

40. 370: The increasing trend in the daily minimum temperature is indeed qualitatively in line with increasing cloudiness, but the latter is very small and insignificant (Table 2). The heat island discussed for the fog trend should have also played a role in the temperature trend.

**A: As mentioned in the previous comment, we discussed the urban heat island effect in the context of temperature trends.**

41. 381: Figure 4: It would be interesting to show how these variables behave for clear skies only. A second set of lines with clear-sky values could be added with different color or symbol.

**A: As the number of clear sky days was too low for a statistically significant analysis, it was removed from the manuscript as pointed by both referees.**

42. 385-390: Please try to split this long sentence in to two. It is difficult to read.

**A: We removed this part of the discussion in the revised version of the manuscript. The effect of aerosol on SSR will be analysed in a future study, when more data related to aerosol properties could be gathered.**

43. 428: Please state the wavelength of the aerosol optical depth data.

**A: Text removed from the revised manuscript.**

44. 442: Does the SSA from AERONET show any trend after 2000? What about the AOD from this instrument?

**A: Text removed from the revised manuscript.**

45. 433: The variability of the AAI and the AOD cannot be compared in absolute terms because these two qualities are not the same.

**A: Text removed from the revised manuscript.**

46. 436: From Figure 5 is evident that there is an abrupt change in AAI between 1992 and 1994 which should not be neglected. The AAI after 1995 has been almost doubled and

remained fairly constant. Considering the years 1984-2016 as one period for a trend is probably not a good choice since the trend is not linear.

**A: Text removed from the revised manuscript.**

47. 470: "their distinct patterns". If I understand correctly, the other factors may have affected the SSR and not the SD and DTR, thus it should be changed to "the distinct changes in SSR".

**A: At this point we meant that other factors may have affected SSR (such as aerosol optical depth, analysed by the variability of visibility data), SD (number of foggy days) and DTR (urban heat island effect and anthropogenic greenhouse gas emissions).**

48. 503-506: See my comment for line 436 above.

**A: Removed from the final version.**

49. Technical

30: Delete "still" – **OK – see line 32.**

34: Replace "encouraged" by "planned" – **OK – see line 37.**

55: Insert "comprising" before "both", and delete the two occurrences of "in" – **OK – see line 59.**

67: Replace "the" by "increasing" – **OK – see line 72.**

92: Replace "propose to answer" with "are addressing" – **OK – see line 98.**

95-97: Better use "section" instead of "part" – **OK – see lines 101 and 102.**

148: Insert "by" after "estimated" – **We modified this part of the text. See lines 171 to 177 of the revised version of the manuscript.**

150: Delete "also" – **OK – see line 174.**

182: Please define that the dry season is July-October. **As previously discussed, the analysis of the aerosol effect was removed from the manuscript.**

242-243: There is a long blank after the P in word Period (two occurences) – **OK – see lines 238 and 239.**

270: Replace "solely" with "sole" – **this sentence was removed from the revised version of the manuscript.**

271: Replace "clue" by "quantitative estimate" – **the text was removed from the revised version of the manuscript**.

275: Please add after "spring" the months corresponding to winter and spring season, just to avoid confusion for the readers living in the northern hemisphere. – **the text was removed from the revised version of the manuscript.**

291: Delete "decade" – **the text was removed from the revised version of the manuscript.**

292: Replace "mention" by "mentioning" – **the text was removed from the revised version of the manuscript.**

293: Replace "n/N" by normalized sunshine duration" – **the text was removed from the revised version of the manuscript.**

331: Delete (FFD) as it is has been already defined – **the text was removed from the revised version of the manuscript.**

333: Please replace "scenarios" with "conditions" – **the text was removed from the revised version of the manuscript.**

339: Replace "decay" with "reduction" since decay usually implies a gradual decrease but here we have a rather abrupt change. – **the text was removed from the revised version of the manuscript.**

357: Replace "it" by "DTR" – **OK – see line 356.**

360: Please add after "space", "during daytime" and after "surface", "during the night" – **OK – see line 359.**

400: Delete "jumping" – **OK – see line 321.**

401: Add "and" before "decreasing" – **OK – see line 322.**

423: Replace "relative" with "relatively" – **the text was removed from the revised version of the manuscript.**

431,432: Something is missing in this sentence. – **the text was removed from the revised version of the manuscript.**

434: Please replace "1980 and 1990 decades" with "in the 1980s and 1990s". – **the text was removed from the revised version of the manuscript.**

471: Delete "a restrict analysis of" – **OK.- see line 418.**

475-476: Delete "is a potential candidate to" – **OK – see line 421.**

477: Replace "Although" with "However," – **the text was removed from the revised version of the manuscript.**

---

## Author Response (AR2)

Editor Decision: Publish subject to minor revisions (review by editor) (10 Mar 2021) by Stelios Kazadzis

Comments to the Author:

Please take into account the second round of comments from the reviewer.

Non-public comments to the Author:

I think the authors have to include the second round of comments from the reviewer.

This is a very interesting time series that has the unique feature of two negative linear series and a "jump" in between.

So I agree with the reviewer that the total change should be mentioned and this sudden change during the 1983 which can be partly linked with cloud changes and with DTR ones have to be explained. Or at least to be mentioned and the authors can provide some explanation that could be revisited in the future.

**Dear Dr. Kazadzis, thank you for your comments. We agree with the comments of the reviewer and changed the manuscript accordingly.**
* * *
**The authors are again grateful to the anonymous referee for taking his/her time reading and making important suggestions in the revised version of the manuscript.**

The present version of the manuscript is much better than the previous version. However, I still have some objections regarding the analysis of the data, and especially the discussion about the results shown in Figure 2.

As I understand from the manuscript, and from the reply of the authors to one of my previous comments, they used the methodology of Rodionov (2004) and detected a statistically significant change in the direction of the trend of DTR in 1983. Based on this result, as well as the results of other studies, they decided to perform the analysis for two sub-periods: 1961 – 1983 and 1983 – 2016.

**Authors reply: Using Rodionov (2004) methodology, based on mean values, the observed regime shift in 1984 was for Tmax, not for DTR. As discussed next, we agree with the reviewer that a piecewise linear regression model should be more appropriate than comparing mean values before and after the shift if it is detected.**

First of all, is 1983 the first, and the only year that the change in the direction of the trend in DTR is significant? For example, I would expect that the change would be significant for a range of 3-4 years around 1983. Some relative discussion should be added in the document. There should also be some similar discussion regarding all parameters shown in Figure 2.

**Authors reply: The change in the trend direction of both SD and DTR was detected using the piecewise linear regression model proposed by Muggeo (2003), as discussed in the previous round of replies to the referees. For SD, the change was detected in 1982 with an uncertainty of 4 years, with p = 0.008 and for DTR, the change was detected in 1979 with an uncertainty of 4 years (p = 0.017). The authors apologize for the lack of clarity and for the incomplete information in the previous reply. The applied model to detect trend changes and respective reference were included in the methodology, at lines 144 to 145: "We also applied a piecewise linear regression model, proposed by Muggeo (2003) to detect any trend changes". The results of the application were included in the revised version of the manuscript, in many parts, as in lines 223 to 228; 231; 237 to 238; 243 to 248; 262 to 264; 382 to 388, including in the Abstract, lines 29 to 34 and Conclusions, lines 430 to 440.**

Secondly, I would recommend to the authors to use a piece-wise linear regression model when they detect a significant change, instead of studying the two trends independently. This way the authors will be able to discuss the trend in e.g. the first period with respect to the trend in the second period. Else, the discussion may be misleading. For example, the discussion about the results for SSR in Figure 2 is misleading. Based on what is discussed in the document (especially in the introduction), the reader would expect that the SSR was decreasing by 0.4% per decade in 1961 – 1983, and then by 0.39% per decade in 1983 – 2016. Thus, an overall decrease of ~2.2% for the whole period (1961 - 2016). This is not of course truth since the SSR is relatively stable during 1961 – 2016.

**Authors reply: As mentioned in the previous comment, a piecewise model was in fact used. And yes, the authors agree with the referee comment about the misleading interpretation given by such numbers, what reinforces the importance of analysing the longest timeseries possible in this kind of study.**

Third, I strongly recommend adding some discussion about the average change throughout the whole period, at least for the parameters for which the regime of the trend is not changing significantly (e.g. SSR).

**Authors reply: The authors agree with the recommendation and included the trends for the whole period for all the analysed variables. The tables 1 (starting in line 249) and 2 (390) incorporate the results.**

In any case, and even if the authors decide not to follow my advice regarding the methodology of the analysis, they should discuss in a much clearer way what has happened in the period of study, and in which cases the discussed results are significant.

I also recommend that the following corrections should be applied.

L23: Replace "up to" with "and" – **replaced.**

L25: Delete "of". Furthermore, the significance level is 89.9%. The p-value is 0.101**. – This part of the text was deleted.**

L26-29: "A similar … 0.013)". Please re-write this sentence because it is not clear.

**Authors reply: In this revised version, the text was rewritten to: "Sunshine duration and the diurnal temperature range also presented negative trends, in opposition to the positive trend observed in the cloud cover fraction".**

L39: "each possible cause" instead of "each possible causes" – **The text was corrected following referee suggestion in line 38.**

L132: "were monitored since" instead of "started to be monitored in" – **The text was changed in the line 131 of the present version of the manuscript.**